# Transcriptomics analysis of the bovine endometrium during the perioestrus period

**Mohammed A. Alfattah**[1,2¤], **Carolina N. Correia**[3], **John A. Browne**[1], **Paul A. McGettigan**[1], **Katarzyna Pluta**[1], **Stephen D. Carrington**[1], **David E. MacHugh**[3,4], **Jane A. Irwin**[1]*

**1** UCD School of Veterinary Medicine, UCD College of Health and Agricultural Sciences, University College Dublin, Belfield, Dublin, Ireland, **2** King Faisal University, Al-Ahsa, Saudi Arabia, **3** Animal Genomics Laboratory, UCD School of Agriculture and Food Science, UCD College of Health and Agricultural Sciences, University College Dublin, Belfield, Dublin, Ireland, **4** UCD Conway Institute of Biomolecular and Biomedical Research, University College Dublin, Belfield, Dublin, Ireland

¤ Current address: Department of Biology, College of Science, Jazan University, Jizan, Saudi Arabia
* jane.irwin@ucd.ie

**Data Availability Statement:** All of the bioinformatics and statistical workflow scripts (Bash, Perl, and R programming languages) used are available from a public GitHub repository

## Abstract

During the oestrous cycle, the bovine endometrium undergoes morphological and functional changes, which are regulated by alterations in the levels of oestrogen and progesterone and consequent changes in gene expression. To clarify these changes before and after oestrus, RNA-seq was used to profile the transcriptome of oestrus-synchronized beef heifers. Endometrial samples were collected from 29 animals, which were slaughtered in six groups beginning 12 h after the withdrawal of intravaginal progesterone releasing devices until seven days post-oestrus onset (luteal phase). The groups represented proestrus, early oestrus, metoestrus and early dioestrus (luteal phase). Changes in gene expression were estimated relative to gene expression at oestrus. Ingenuity Pathway Analysis (IPA) was used to identify canonical pathways and functional processes of biological importance. A total of 5,845 differentially expressed genes (DEGs) were identified. The lowest number of DEGs was observed at the 12 h post-oestrus time point, whereas the greatest number was observed at Day 7 post-oestrus onset (luteal phase). A total of 2,748 DEGs at this time point did not overlap with any other time points. Prior to oestrus, *Neurological disease* and *Organismal injury and abnormalities* appeared among the top IPA diseases and functions categories, with upregulation of genes involved in neurogenesis. Lipid metabolism was upregulated before oestrus and downregulated at 48h post-oestrus, at which point an upregulation of immune-related pathways was observed. In contrast, in the luteal phase the *Lipid metabolism* and *Small molecule biochemistry pathways* were upregulated.

## Introduction

In female mammals, the uterus is an organ that serves several purposes, including sperm transport, implantation of the conceptus, placentation, and the maintenance of pregnancy. During the oestrous cycle, levels of steroid hormones fluctuate, permitting reproductive receptivity, and enabling successful establishment of pregnancy. This cycle facilitates the physiological

https://github.com/carolcorreia/Estrus-Endometrium-RNA-sequencing All RNA-Seq raw data used in this paper are available from the European Nucleotide Archive (ENA, https://www.ebi.ac.uk/ena/browser/search ) with accession number PRJEB33671.

**Funding:** This work was supported by a grant to MAA from the Government of Saudi Arabia (No. IR12012/2 and IR1610) and a Brazilian Science Without Borders – CAPES grant (No: BEX-13070-13-4) to CNC. The funders had no role in study design, data collection and analysis, decision to publish, or preparation of the manuscript.

**Competing interests:** The authors have declared that no competing interests exist.

remodelling of the endometrium and prepares it for embryonic implantation should conception occur [1]. In cattle, this 21-day cycle is divided into four phases: proestrus, oestrus, metoestrus and dioestrus [2,3], and the morphological changes which occur are regulated by both progesterone (P4) and oestrogen (E2). Proestrus occurs when circulating P4 levels are at basal level and luteinizing hormone (LH) pulse frequency and E2 concentrations increase and peak. The latter peaks just before ovulation (Day 0) at which time the animal is reproductively receptive [4]. As the animal enters metoestrus, the corpus luteum begins to develop, leading to an increase in P4, which peaks about day 12 (dioestrus). During this process, the endometrium transforms from a proliferative to a secretory state [5]. In the secretory state, uterine gland secretion is at maximum until implantation begins, producing 'histotroph' that contains many nutrients essential for successful implantation and nourishment of the conceptus and its interactions with the mother [6,7].

Around the time of oestrus, spermatozoa introduced into the cranial vagina enter the cervical canal on their way to the uterus and oviduct, which contains mucus-filled micro-grooves that conduct sperm into the uterus and from there to the oviduct [8]. Some sperm are retained by uterine glands [9], while the rest are transported towards the oviduct by smooth muscle contractions [10]. The transport process appears to occur faster in the late follicular phase, suggesting that the endocrine environment and consequent changes in gene expression are also important to enable this process [11]. Since sperm cells are immunologically foreign, and mating introduces bacteria into the uterus, a local inflammatory response occurs that removes the excess cells and debris. This includes a rapid influx of polymorphonuclear (PMN) leukocytes along with the activation of the adaptive immune response [12]. Seminal fluid can also modulate inflammatory responses in the female reproductive tract that enhance the possibility of fertilization and pregnancy [13] and PMN cells phagocytose the sperm, inducing formation of neutrophil extracellular traps [14].

The endometrium undergoes significant change to facilitate sperm transport and removal, along with the transition from a proliferative to a secretory state, and these are reflected in the transcriptome. In the proliferative state, the endometrium thickens, with an increase in vascularization and mitosis in stromal and epithelial cells, along with increased ciliation of the epithelium. Cell proliferation decreases in the secretory stage, while endometrial glands increase in size [15]. These changes ensure that the maternal environment is optimal for supporting the embryo post-implantation and these physiological changes will be reflected in the transcriptome. The endometrial transcriptome is regulated by endocrine and species-specific factors [16], including the periovulatory endocrine environment, particularly the size of the preovulatory follicle [17], and postovulatory P4 concentrations [18–21]. Other external factors such as uterine disease [22–26] and genetic merit for fertility [27,28] contribute to this highly regulated process. Following implantation, further changes occur at a transcriptomic level in response to the presence of a conceptus [29–31], and proximity to the conceptus [32]. The endometrial transcriptome was found to differ at 6–8 days post-oestrus between cows that became pregnant after embryo transfer and those that did not do so, particularly with respect to genes regulating histotroph composition and endometrial receptivity [33]. The location in relation to the corpus luteum also affected the bovine endometrial response to the conceptus [34]. One study [35] showed that when ovariectomized cows were treated with E2, P4, or a combination of both, to determine how these hormones affected the transcriptome, separate clusters of genes were regulated by E2. These included genes involved in the cell cycle, morphogenesis, and differentiation, whereas the profile of the P4-regulated clusters included genes involved in luteinization, oocyte maturation, and catecholamine metabolism. Furthermore, components of seminal plasma were shown to alter the endometrial transcriptome [36]. Artificial insemination of heifers with semen from low and high-fertility bulls led to differential gene expression,

with high-fertility semen leading to increased expression of genes involved in immune response and inflammation [37]. Transcriptomic studies have also identified signalling molecules supporting embryonic development in the first seven days after ovulation [38]. In addition, the endometrial microbiota is associated with differential gene expression in postpartum cows, and this variation may be modulated by a pathway that works by affecting ovarian cyclicity on the endometrium [39].

There have been relatively few transcriptomic studies of the endometrium concentrating specifically on the perioestrus period in bovines. Most to date concentrate on the luteal phase, implantation, pregnancy, and the post-partum period. The current study uses endometrial tissues harvested in tandem with endocervical tissues derived from oestrus-synchronized heifers. The data on the cervical transcriptome of these animals have been published previously by our group [40,41] and also later analysis of this dataset was performed by Gonçalves et al. [42], who compared the differentially expressed genes (DEGs) for the follicular and luteal phases. In this study, RNA-sequencing (RNA-seq) data was generated from samples derived from the endometrial tissue of oestrus-synchronized heifers during the perioestrus period, allowing investigation of changes in the transcriptome as the endometrium was exposed to changes in E2 and elevation in P4 concentrations, with the final sample taken at Day 7 after the onset of oestrus (early luteal phase).

We hypothesized that the endometrium undergoes changes in gene expression around the time of oestrus in response to fluctuations in E2 and P4 concentrations, and that these changes are related to the facilitation of sperm transport towards the oviduct and the subsequent preparation of the endometrium for successful embryonic implantation. The aim of this study was to generate and compare gene expression profiles at five perioestrus time points against the onset of oestrus, and these have been related in this work to the physiological changes that occur just before and after oestrus.

## Materials and methods

### Animals, reproductive management, and tissue collection

Twenty-nine nulliparous mixed-breed beef heifers were used, as described by Pluta et al. [40] who used cervical tissue from the same animals for transcriptomic analysis. All procedures were licensed by the Irish Department of Health and Children in accordance with European Community Directive 86/609/EC and the Cruelty to Animals Act (Ireland, 1876) and were carried out with approval of UCD Animal Research Ethics Committee (permit ID: AREC-P-08-36-Pluta-Carrington).

To synchronize oestrus, a controlled internal drug release device (CIDR, Zoetis, Dublin, Ireland) was inserted for 8 days. The day before CIDR removal the cattle were injected intramuscularly with 2 ml (0.25 mg/ml) of a synthetic prostaglandin analogue, (PGF2α-Estrumate, Chanelle Pharmaceuticals, Loughrea, Ireland). Heifers were checked for signs of oestrus every 6 h post-CIDR removal until 60 h afterwards. All animals came into oestrus between 48–72 h post-CIDR removal [40]. Blood samples were taken by jugular vein puncture every 12 h on Day 0 (the day CIDR was removed), every 6 h on Day 1, every 3 h on Day 2, every 3 h on Day 3, every 12 h on Day 4, and once a day from Day 5 to Day 8. Serum was separated from plasma within 12 h of sampling and was stored at −20˚C. LH, E2 and P4 concentrations were measured according to the methods of Cooke et al. [43] and Forde et al. [20]. Serum concentrations of E2, P4 and LH for the animals taken at different time points have been published previously [40].

Endometrial tissue samples were collected from the heifers after slaughter at a local licensed abattoir as follows: Group 1: 12 h post- CIDR removal (n = 6); Group 2: 24 h post- CIDR

removal (n = 6); Group 3: at the onset of oestrus (n = 4); Group 4: 12 h post-oestrus onset (n = 4); Group 5: 48 h post-oestrus onset (n = 4); and Group 6: Day 7 post-oestrus onset (n = 5) (luteal phase). Reproductive tracts were opened longitudinally. Both caruncular and intercaruncular tissues were sampled. Endometrial mucosa, which included all cell types, was removed and placed in RNAlater™ (Thermo Fisher Scientific, Dublin, Ireland), then transferred to microcentrifuge tubes after 24 h and stored at −80˚C.

## RNA preparation

The samples stored at -80˚C were used for RNA preparation. This work was performed separately to that of Pluta *et al.* [40]. Total RNA was isolated from endometrial homogenate using TRIzol® reagent (Thermo Fisher Scientific, Dublin, Ireland) according to the manufacturer's instructions. Endometrial tissue (50 mg) was homogenized in 1 ml of TRIzol® using stainless steel beads and a Qiagen (Crawley, U.K.) TissueLyzer II (2 × 120 sec, maximum speed). After homogenization, 200 μl of chloroform was added to each sample, centrifuged (12,000 × *g*, 15 min), and the upper aqueous phase was transferred to a clean micro-centrifuge tube. RNA was precipitated using an equal volume of isopropanol. The RNA was purified using a Qiagen RNeasy mini kit (Qiagen, Crawley, UK) according to the manufacturer's recommendations.

A NanoDrop spectrophotometer ND-1000 (Nanodrop, Denver, USA) was used to quantify RNA concentration and an Agilent Bioanalyzer 2100 (Agilent Technologies, Santa Clara, USA) was used to confirm RNA quality using an RNA Nano chip. The 260/280 absorbance ratio ranged between 1.92 and 2.18 for all samples. The RNA integrity number (RIN) was greater than 7.5 for all samples with an average value of 9.2. The rRNA ratio (28S/18S) varied from 1.1 and 2 between individual samples. These data are provided in S1 Table.

## RNA-seq library preparation and sequencing

RNA-seq library preparation and sequencing were performed by BGI Genomics, Shenzhen, China. This work was carried out separately to that of Pluta *et al.* [40]. Briefly, mRNA was isolated from 200 ng total RNA and purified by oligo-dT beads, and the TruSeq RNA Sample Prep Kit v2 (Illumina, San Diego, USA) protocol was used to construct the libraries. The RNA was fragmented with Elute, Prime, Fragment Mix (EPF, Illumina). First-strand cDNA synthesis was carried out using First Strand Master Mix and Super Script II reverse transcription (Invitrogen). The reaction conditions were 25˚C for 10 min, 42˚C for 50 min, and 70˚C for 15 min. The product was purified using RNAClean XP Beads (Beckman Coulter Life Sciences, Indianapolis, USA), followed by addition of second strand Master Mix and dATP, dGTP, dCTP, dUTP mix to synthesize the second-strand cDNA at 16˚C for 1 h. To end repair, a single dATP nucleotide was added to each strand, and adaptor ligation was performed. The purified fragmented cDNAs combined with End-Repair Mix were incubated for 30 min at 30˚C. The end-repaired DNA was purified with AMPure XP Beads (Beckman Coulter Life Sciences), and A-Tailing Mix was added and incubated for 30 min at 37˚C after mixing. Adapter ligation was performed by adding adenylated 3' end DNA, RNA Index Adapter and Ligation Mix, mixing well by pipetting, then incubating for 10 min at 30˚C. The end-repaired DNA was purified with AMPure XP Beads.

Uracil-N-glycosylase was added, and the mixture was incubated for 10 min at 37˚C, followed by product purification with AMPure XP Beads. Polymerase chain reaction amplification with PCR Primer Cocktail and PCR Master Mix (Illumina) was performed to enrich the cDNA fragments. The PCR products were purified with AMPure XP Beads. The libraries were validated by measuring the distribution of the fragment size using an Agilent 2100 Bioanalyser

(Agilent Technologies) and by assessing library quantification through RT-qPCR with Taq-Man Probe (Thermo Fisher Scientific).

The libraries were then amplified on a cBot system (Illumina, San Diego, USA). The cluster was generated on a flowcell (TruSeq PE Cluster Kit V3–cBot–HS, Illumina). A total of 29 paired-end 2 × 90 bp libraries were sequenced with the HiSeq 2000 System (TruSeq SBS KIT-HS V3, Illumina).

## Bioinformatics and statistical analysis of RNA-seq differential gene expression

Bioinformatics procedures were performed on a 32-core Linux Compute Server (4 × AMD Opteron™ 6220 processors at 3.0 GHz with 8 cores each), with 256 GB of RAM, 24 TB of hard disk storage, and with Ubuntu Linux OS (version 14.04.4 LTS). All of the bioinformatics and statistical workflow scripts (Bash, Perl, and R programming languages) used in this work are available from a public GitHub repository (https://github.com/carolcorreia/Oestrus-Endometrium-RNA-sequencing). Fig 1 shows the RNA-seq bioinformatics and statistical workflows. Sequence data files for the 29 RNA-seq libraries have been deposited in the European Nucleotide Archive (ENA, https://www.ebi.ac.uk/ena/browser/search) with accession number PRJEB33671.

Filtering and segregation of sequencing reads based on the unique RNA-seq library barcode index sequences were performed by BGI Genomics using a pipeline that simultaneously demultiplexed and converted pooled sequence reads to discrete FASTQ files for each RNA-seq sample with no barcode index mismatches permitted. The quality of the individual 29 raw RNA-seq sample library files was assessed with the FastQC software (version 0.11.5) [44]. Paired-end reads from each individual library were aligned to the *Bos taurus* reference genome (UMD3.1.1 –GCF_000003055.6; [45,46], from the NCBI RefSeq database [47], using the STAR aligner software (version 2.5.1b) [48]. Following this, the aligned SAM files were quality-assessed with FastQC. For each library, raw counts for each gene were obtained using the featureCounts software from the Subread package (version 1.5.1-p1) [49], with parameters set to

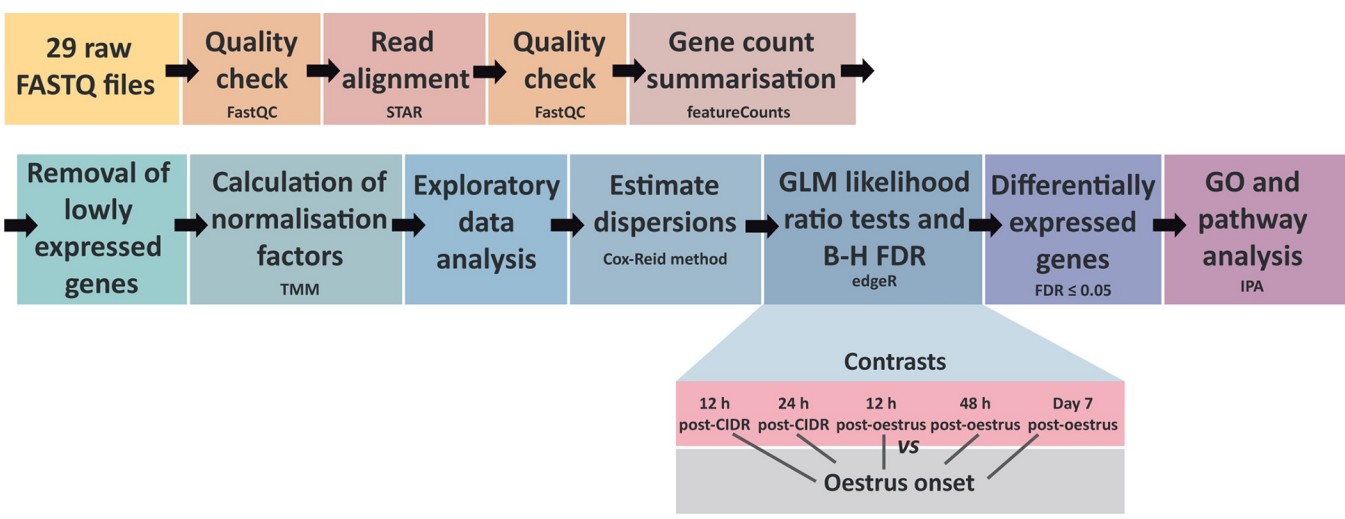

**Fig 1. Diagram showing RNA-seq bioinformatics and statistical analysis.** The main workflow steps and their order are shown. These computational workflows are available at https://github.com/carolcorreia/Estrus-Endometrium-RNA-sequencing.

assign unambiguously uniquely aligned paired-end reads in a stranded manner to the exons of genes within the UMD 3.1.1 *B. taurus* reference genome annotation (NCBI GCF_000003055.6 –genomic GFF file).

Raw gene count outputs from featureCounts were used to establish DEG analysis within an R-based workflow (RStudio IDE version 1.0.136; [50]), running R version 3.3.0 [51] using the edgeR package (version 3.14.0) [52] from the Bioconductor project (version 3.4, with BiocInstaller 1.24.0) [53]. The org.Bt.eg.db annotation package (version 3.3.0) was used to obtain gene names and symbols [54]. Genes displaying expression levels below the minimally set threshold of one count per million (CPM) in at least four individual libraries (i.e., equivalent to the smallest group of biological replicates) were discarded from downstream analysis. Normalization factors were calculated for each library using the trimmed mean of M-values method [55]. The Cox-Reid method was used to estimate the dispersion parameter for each library [56]. DEGs between the perioestrus groups (12 h post-CIDR removal, 24 h post-CIDR removal, 12 h post-oestrus, 48 h post-oestrus, and Day 7 of the cycle) versus the oestrus onset group (i.e., unpaired-sample statistical model) were then identified using a negative binomial generalized linear model. Multiple testing correction was done using the Benjamini–Hochberg (B-H) method [57] with a false discovery rate (FDR) $\leq 0.05$. Data visualization was performed with the R packages ggplot2 (version 2.2.1) [58] and VennDiagram (version 1.6.17) [59].

## Pathway analyses of differentially expressed genes

Ingenuity® Pathway Analysis (IPA) (Qiagen, Redwood City, CA, USA) software package (version 01–07) [60] was used to identify overrepresented (enriched) disease and functions categories, canonical pathways, interaction networks, and upstream regulators for sets of DEGs at each perioestrus time point (12 h post-CIDR removal, 24 h post- CIDR removal, 12 h post-oestrus, 48 h post-oestrus, and Day 7 post-oestrus (luteal phase) compared to the onset of oestrus time point.

## Results

### RNA-seq alignment and summary statistics

Individually barcoded strand-specific RNA-seq libraries generated from *B. taurus* endometrial tissue were obtained from 29 animals at six time points. Summary statistics relating to each individual library are provided in S2 Table. Alignment of the paired-end reads to the *B. taurus* reference genome (UMD 3.1.1) yielded mean values per library of 23,968,186 ± 900,849 reads (n = 29 libraries, ± standard deviation) corresponding to a mean of 95.6% reads mapping to unique locations in the bovine genome. A mean of 655,128 ± 45,674 reads (2.6%) mapped to multiple locations in the genome, and a mean of 34,158 ± 8,568 reads (0.14%) multi-mapped to more than 10 locations in the genome. Finally, a mean of 416,755 ± 58,839 reads (1.66%) failed to map to any genomic location. The mean mapped length was 179.2 ± 0.15 bp.

A mean value of 21,327,606 reads (80.3%) were assigned to annotated regions of the *B. taurus* genome, and 153,681 ± 13,805 reads (0.58%) were assigned to ambiguous or overlapping annotated genomic regions. Unassigned no features reads comprised 9.3%, or 2,486,900 ± 787,328, while there were 2,611,535 ± 326,596 (9.84%) unassigned multi-mapping reads. For the 29 RNA-seq libraries, filtering of genes expressed at a low level resulted in 16,922 genes of a total of 24,616 annotated *B. taurus* genes being suitable for differential expression analysis.

### Summary of differential gene expression of RNA-seq data

Gene expression for the five different groups was compared to that at the oestrus time point (Fig 2A). The total number of DEGs at the five time points examined relative to oestrus was

**A)**

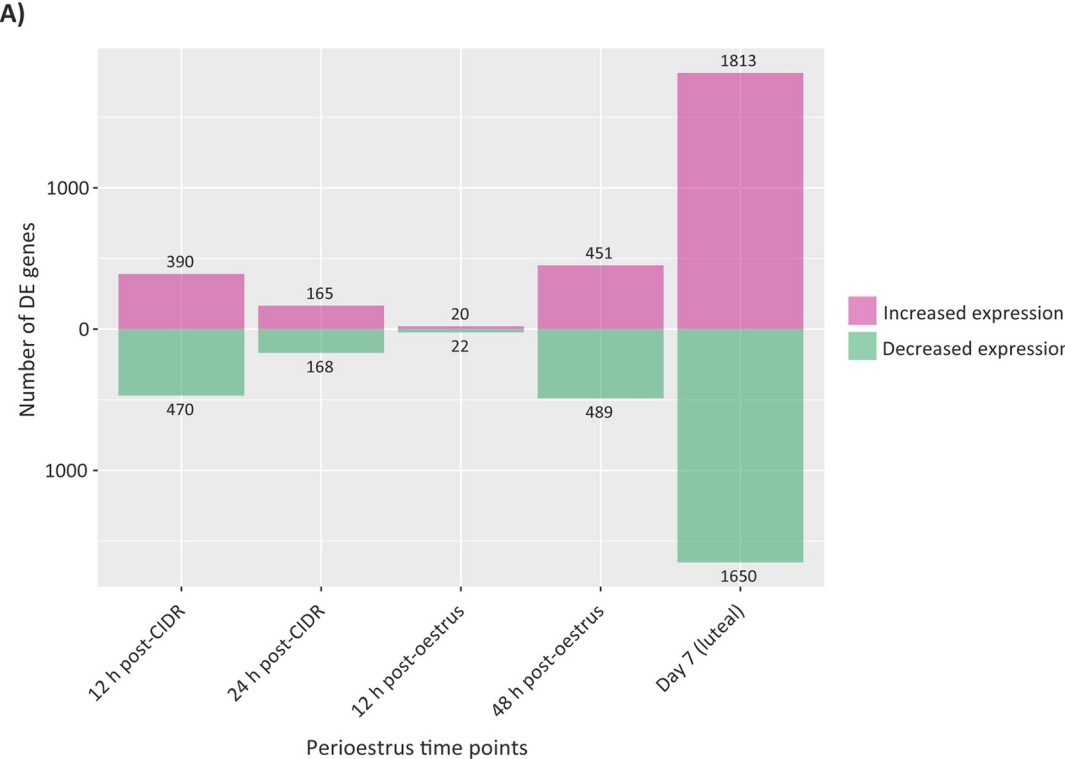

**B)**

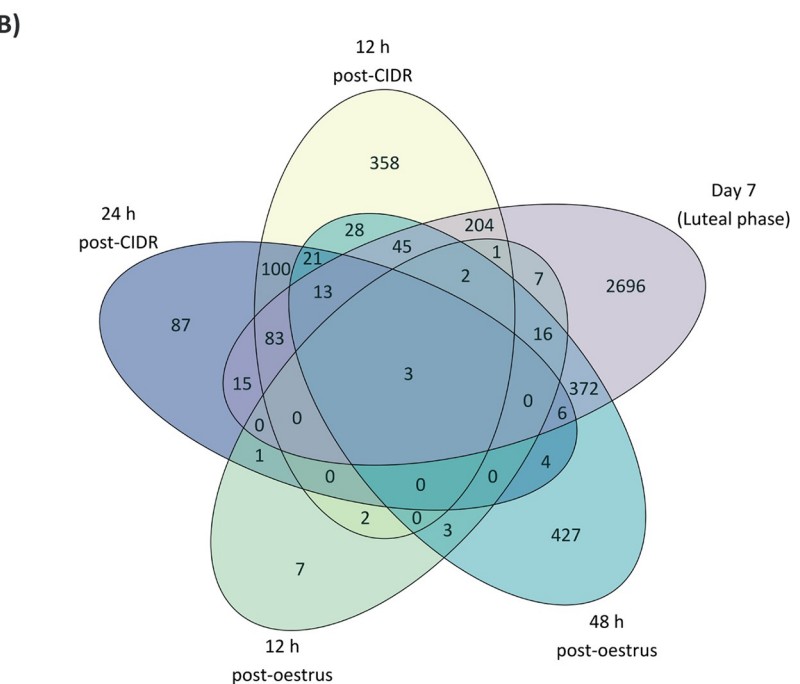

**Fig 2. Visualisation of significantly differentially expressed genes (DEGs).** A) The number of DEGs with increased and decreased expression at each perioestrus time point, relative to expression at the oestrus time point is shown. B) Venn diagram of numbers of DEGs for the five perioestrus time points relative to oestrus.

**Table 1. Top DEGs in bovine endometrium at 12 h post-CIDR removal versus oestrus, ranked by ascending B-H FDR.**

| Entrez ID | Gene Name | Gene Symbol | Log$_2$ fold-change | P-Value | B-H FDR |
|---|---|---|---|---|---|
| **Upregulated** | | | | | |
| 540267 | ST8 alpha-N-acetyl-neuraminide alpha-2,8-sialyltransferase 2 | ST8SIA2 | 2.05 | 1.90×10$^{-08}$ | 1.53×10$^{-05}$ |
| 504960 | CD68 molecule | CD68 | 1.93 | 2.30×10$^{-08}$ | 1.70×10$^{-05}$ |
| 514833 | C-type lectin domain family 10, member A | CLEC10A | 4.66 | 4.47×10$^{-08}$ | 2.77×10$^{-05}$ |
| 521133 | SEC14-like 3 (S. cerevisiae) | SEC14L3 | 5.82 | 4.25×10$^{-08}$ | 2.77×10$^{-05}$ |
| 527114 | angiotensinogen (serpin peptidase inhibitor, clade A, member 8) | AGT | 4.83 | 4.00×10$^{-08}$ | 2.77×10$^{-05}$ |
| 751788 | WC1 isolate CH210 | LOC751788 | 3.81 | 4.40×10$^{-08}$ | 2.77×10$^{-05}$ |
| 527903 | V-set and transmembrane domain containing 2 like | VSTM2L | 2.65 | 6.38×10$^{-08}$ | 3.60×10$^{-05}$ |
| 790164 | SLAM family member 7 | SLAMF7 | 2.84 | 1.27×10$^{-07}$ | 6.35×10$^{-05}$ |
| 613715 | chromosome 28 open reading frame, human C10orf10 | C28H10orf10 | 3.09 | 1.66×10$^{-07}$ | 7.22×10$^{-05}$ |
| 617478 | coiled-coil domain-containing protein C1orf110 homolog | C3H1orf110 | 2.98 | 2.27×10$^{-07}$ | 9.37×10$^{-05}$ |
| **Downregulated** | | | | | |
| 101906058 | acyl-CoA desaturase-like | LOC101906058 | -3.03 | 2.06×10$^{-14}$ | 3.48×10$^{-10}$ |
| 504813 | neuropeptide Y receptor Y1 | NPY1R | -2.62 | 2.83×10$^{-13}$ | 2.05×10$^{-09}$ |
| 509003 | nucleobindin 2 | NUCB2 | -2.28 | 3.63×10$^{-13}$ | 2.05×10$^{-09}$ |
| 280924 | stearoyl-CoA desaturase (delta-9-desaturase) | SCD | -2.8 | 1.07×10$^{-11}$ | 4.53×10$^{-08}$ |
| 534049 | family with sequence similarity 213, member A | FAM213A | -2.77 | 8.45×10$^{-11}$ | 2.86×10$^{-07}$ |
| 767936 | purinergic receptor P2Y, G-protein coupled, 14 | P2RY14 | -2.41 | 2.78×10$^{-10}$ | 7.85×10$^{-07}$ |
| 281356 | natriuretic peptide C | NPPC | -3.69 | 4.59×10$^{-10}$ | 8.62×10$^{-07}$ |
| 286767 | parathyroid hormone-like hormone | PTHLH | -3.08 | 4.13×10$^{-10}$ | 8.62×10$^{-07}$ |
| 538255 | hyperpolarization activated cyclic nucleotide-gated potassium channel 1 | HCN1 | -4.38 | 4.33×10$^{-10}$ | 8.62×10$^{-07}$ |
| 407767 | 3-hydroxy-3-methylglutaryl-CoA synthase 1 (soluble) | HMGCS1 | -2.24 | 8.65×10$^{-10}$ | 1.46×10$^{-06}$ |

5,845 out of a total of 16,922 *B. taurus* genes (34.5%) annotated in RefSeq. Statistical analysis of the data using edgeR with a B-H FDR-adjusted $P$ ($P_{adj}$) $\leq$ 0.05 identified 860 DEGs at 12 h post-CIDR removal and 333 at 24 h post-CIDR removal. The 12 h post-oestrus time point, which is closest temporally and in terms of hormone levels to oestrus, gave rise to the lowest number of DEGs, with only 42 DEGs. At 48 h post-oestrus 940 DEGs were observed and the largest increase in DEGs was present in the Day 7 post-oestrus sample, which had 3,463 DEGs. Fig 2B revealed a total of 2,696 of the DEGs out of a total of 3,463 that are unique to the luteal phase and do not overlap with any of the other five time points, whereas only seven genes were uniquely differentially expressed at 12 h post-oestrus.

Tables 1–5 list the top upregulated and downregulated genes between each of the perioestrus groups in comparison to oestrus. A complete list of DEGs at each time point analysed is provided in S3 Table.

Several of the genes in each category were defined as uncharacterized. For example, two of the ten upregulated genes in luteal phase versus oestrus (Table 5) are defined as uncharacterized and displayed high log$_2$ fold changes. The majority of the uncharacterized genes listed in Tables 1 to 5 are defined in the latest reference sequence of the bovine genome (ARS-UCD1.2; [61]) as non-coding RNAs, with the exception of *LOC104972601*, *LOC104973104*, and *LOC104973411*, which were withdrawn by NCBI, as the model on which they were based was not predicted in a later annotation. Of the total DEGs, only three showed statistically significant differences in expression for each of the five time points versus oestrus. These were collagen type VI alpha 6 (*COL6A6*), mucin-6 (*LOC101903030*) and angiotensinogen (*AGT*).

**Table 2. Top DEGs in bovine endometrium at 24 h post-CIDR removal versus oestrus, ranked by ascending B-H FDR.**

| Entrez ID | Gene Name | Gene Symbol | Log$_2$ fold-change | P-Value | B-H FDR |
|---|---|---|---|---|---|
| **Upregulated** | | | | | |
| 540267 | ST8 alpha-N-acetyl-neuraminide alpha-2,8-sialyltransferase 2 | ST8SIA2 | 2.05 | $1.90\times10^{-08}$ | $1.53\times10^{-05}$ |
| 504960 | CD68 molecule | CD68 | 1.93 | $2.30\times10^{-08}$ | $1.70\times10^{-05}$ |
| 514833 | C-type lectin domain family 10, member A | CLEC10A | 4.66 | $4.47\times10^{-08}$ | $2.77\times10^{-05}$ |
| 521133 | SEC14-like 3 (S. cerevisiae) | SEC14L3 | 5.82 | $4.25\times10^{-08}$ | $2.77\times10^{-05}$ |
| 527114 | angiotensinogen (serpin peptidase inhibitor, clade A, member 8) | AGT | 4.83 | $4.00\times10^{-08}$ | $2.77\times10^{-05}$ |
| 751788 | WC1 isolate CH210 | LOC751788 | 3.81 | $4.40\times10^{-08}$ | $2.77\times10^{-05}$ |
| 527903 | V-set and transmembrane domain containing 2 like | VSTM2L | 2.65 | $6.38\times10^{-08}$ | $3.60\times10^{-05}$ |
| 790164 | SLAM family member 7 | SLAMF7 | 2.84 | $1.27\times10^{-07}$ | $6.35\times10^{-05}$ |
| 613715 | chromosome 28 open reading frame, human C10orf10 | C28H10orf10 | 3.09 | $1.66\times10^{-07}$ | $7.22\times10^{-05}$ |
| 617478 | coiled-coil domain-containing protein C1orf110 homolog | C3H1orf110 | 2.98 | $2.27\times10^{-07}$ | $9.37\times10^{-05}$ |
| **Downregulated** | | | | | |
| 101906058 | acyl-CoA desaturase-like | LOC101906058 | -3.03 | $2.06\times10^{-14}$ | $3.48\times10^{-10}$ |
| 504813 | neuropeptide Y receptor Y1 | NPY1R | -2.62 | $2.83\times10^{-13}$ | $2.05\times10^{-09}$ |
| 509003 | nucleobindin 2 | NUCB2 | -2.28 | $3.63\times10^{-13}$ | $2.05\times10^{-09}$ |
| 280924 | stearoyl-CoA desaturase (delta-9-desaturase) | SCD | -2.8 | $1.07\times10^{-11}$ | $4.53\times10^{-08}$ |
| 534049 | family with sequence similarity 213, member A | FAM213A | -2.77 | $8.45\times10^{-11}$ | $2.86\times10^{-07}$ |
| 767936 | purinergic receptor P2Y, G-protein coupled, 14 | P2RY14 | -2.41 | $2.78\times10^{-10}$ | $7.85\times10^{-07}$ |
| 281356 | natriuretic peptide C | NPPC | -3.69 | $4.59\times10^{-10}$ | $8.62\times10^{-07}$ |
| 286767 | parathyroid hormone-like hormone | PTHLH | -3.08 | $4.13\times10^{-10}$ | $8.62\times10^{-07}$ |
| 538255 | hyperpolarization activated cyclic nucleotide-gated potassium channel 1 | HCN1 | -4.38 | $4.33\times10^{-10}$ | $8.62\times10^{-07}$ |
| 407767 | 3-hydroxy-3-methylglutaryl-CoA synthase 1 (soluble) | HMGCS1 | -2.24 | $8.65\times10^{-10}$ | $1.46\times10^{-06}$ |

**Table 3. Top DEGs in bovine endometrium at 12 h post-oestrus versus oestrus, ranked by ascending B-H FDR.**

| Entrez ID | Gene Name | Gene Symbol | Log$_2$ fold-change | P-Value | B-H FDR |
|---|---|---|---|---|---|
| **Upregulated** | | | | | |
| 282139 | arachidonate 15-lipoxygenase | ALOX15 | 3.44 | $1.25\times10^{-07}$ | $2.12\times10^{-03}$ |
| 100297044 | C-C motif chemokine 14 | LOC100297044 | 7.94 | $2.61\times10^{-07}$ | $2.21\times10^{-03}$ |
| 527872 | potassium voltage-gated channel, subfamily H (eag-related), member 6 | KCNH6 | 4.1 | $1.48\times10^{-06}$ | $6.23\times10^{-03}$ |
| 509102 | pipecolic acid oxidase | PIPOX | 4.61 | $7.09\times10^{-06}$ | $1.20\times10^{-02}$ |
| 525276 | ADAM metallopeptidase with thrombospondin type 1 motif, 12 | ADAMTS12 | 2.03 | $6.63\times10^{-06}$ | $1.20\times10^{-02}$ |
| 530102 | collagen, type VI, alpha 6 | COL6A6 | 6.94 | $6.13\times10^{-06}$ | $1.20\times10^{-02}$ |
| 515988 | signal transducer and activator of transcription 4 | STAT4 | 2.34 | $1.53\times10^{-05}$ | $1.85\times10^{-02}$ |
| 101909328 | uncharacterized LOC101909328 | LOC101909328 | 2.1 | $1.77\times10^{-05}$ | $1.87\times10^{-02}$ |
| 534439 | PDZ domain containing 1 | PDZK1 | 2.64 | $1.95\times10^{-05}$ | $1.94\times10^{-02}$ |
| 513680 | NHS-like 2 | NHSL2 | 1.65 | $2.50\times10^{-05}$ | $2.35\times10^{-02}$ |
| **Downregulated** | | | | | |
| 518974 | ATPase, H+ transporting, lysosomal V0 subunit a4 | ATP6V0A4 | -4.34 | $5.56\times10^{-07}$ | $3.13\times10^{-03}$ |
| 101903030 | mucin-6 | LOC101903030 | -6.57 | $1.84\times10^{-06}$ | $6.23\times10^{-03}$ |
| 516256 | tumor necrosis factor receptor superfamily, member 18 | TNFRSF18 | -4.41 | $2.59\times10^{-06}$ | $7.32\times10^{-03}$ |
| 514039 | epithelial cell adhesion molecule | EPCAM | -1.18 | $6.66\times10^{-06}$ | $1.20\times10^{-02}$ |
| 510137 | transmembrane protein 213 | TMEM213 | -7.09 | $1.28\times10^{-05}$ | $1.81\times10^{-02}$ |
| 539063 | carbohydrate (N-acetylglucosamine 6-O) sulfotransferase 4 | CHST4 | -7.51 | $1.39\times10^{-05}$ | $1.81\times10^{-02}$ |
| 785366 | embigin | EMB | -2.83 | $1.34\times10^{-05}$ | $1.81\times10^{-02}$ |
| 535458 | polypeptide N-acetylgalactosaminyltransferase 3 | GALNT3 | -2.28 | $1.72\times10^{-05}$ | $1.87\times10^{-02}$ |
| 540234 | family with sequence similarity 46, member A | FAM46A | -2.55 | $3.13\times10^{-05}$ | $2.79\times10^{-02}$ |
| 404111 | keratin 14 | KRT14 | -5.87 | $3.72\times10^{-05}$ | $3.00\times10^{-02}$ |

**Table 4. Top DEGs in bovine endometrium at 48 h post-oestrus versus oestrus, ranked by ascending B-H FDR.**

| Entrez ID | Gene Name | Gene Symbol | Log$_2$ fold-change | P-Value | B-H FDR |
|---|---|---|---|---|---|
| **Upregulated** | | | | | |
| 540267 | ST8 alpha-N-acetyl-neuraminide alpha-2,8-sialyltransferase 2 | ST8SIA2 | 2.05 | $1.90 \times 10^{-08}$ | $1.53 \times 10^{-05}$ |
| 504960 | CD68 molecule | CD68 | 1.93 | $2.30 \times 10^{-08}$ | $1.70 \times 10^{-05}$ |
| 514833 | C-type lectin domain family 10, member A | CLEC10A | 4.66 | $4.47 \times 10^{-08}$ | $2.77 \times 10^{-05}$ |
| 521133 | SEC14-like 3 (S. cerevisiae) | SEC14L3 | 5.82 | $4.25 \times 10^{-08}$ | $2.77 \times 10^{-05}$ |
| 527114 | angiotensinogen (serpin peptidase inhibitor, clade A, member 8) | AGT | 4.83 | $4.00 \times 10^{-08}$ | $2.77 \times 10^{-05}$ |
| 751788 | WC1 isolate CH210 | LOC751788 | 3.81 | $4.40 \times 10^{-08}$ | $2.77 \times 10^{-05}$ |
| 527903 | V-set and transmembrane domain containing 2 like | VSTM2L | 2.65 | $6.38 \times 10^{-08}$ | $3.60 \times 10^{-05}$ |
| 790164 | SLAM family member 7 | SLAMF7 | 2.84 | $1.27 \times 10^{-07}$ | $6.35 \times 10^{-05}$ |
| 613715 | chromosome 28 open reading frame, human C10orf10 | C28H10orf10 | 3.09 | $1.66 \times 10^{-07}$ | $7.22 \times 10^{-05}$ |
| 617478 | coiled-coil domain-containing protein C1orf110 homolog | C3H1orf110 | 2.98 | $2.27 \times 10^{-07}$ | $9.37 \times 10^{-05}$ |
| **Downregulated** | | | | | |
| 101906058 | acyl-CoA desaturase-like | LOC101906058 | -3.03 | $2.06 \times 10^{-14}$ | $3.48 \times 10^{-10}$ |
| 504813 | neuropeptide Y receptor Y1 | NPY1R | -2.62 | $2.83 \times 10^{-13}$ | $2.05 \times 10^{-09}$ |
| 509003 | nucleobindin 2 | NUCB2 | -2.28 | $3.63 \times 10^{-13}$ | $2.05 \times 10^{-09}$ |
| 280924 | stearoyl-CoA desaturase (delta-9-desaturase) | SCD | -2.8 | $1.07 \times 10^{-11}$ | $4.53 \times 10^{-08}$ |
| 534049 | family with sequence similarity 213, member A | FAM213A | -2.77 | $8.45 \times 10^{-11}$ | $2.86 \times 10^{-07}$ |
| 767936 | purinergic receptor P2Y, G-protein coupled, 14 | P2RY14 | -2.41 | $2.78 \times 10^{-10}$ | $7.85 \times 10^{-07}$ |
| 281356 | natriuretic peptide C | NPPC | -3.69 | $4.59 \times 10^{-10}$ | $8.62 \times 10^{-07}$ |
| 286767 | parathyroid hormone-like hormone | PTHLH | -3.08 | $4.13 \times 10^{-10}$ | $8.62 \times 10^{-07}$ |
| 538255 | hyperpolarization activated cyclic nucleotide-gated potassium channel 1 | HCN1 | -4.38 | $4.33 \times 10^{-10}$ | $8.62 \times 10^{-07}$ |
| 407767 | 3-hydroxy-3-methylglutaryl-CoA synthase 1 (soluble) | HMGCS1 | -2.24 | $8.65 \times 10^{-10}$ | $1.46 \times 10^{-06}$ |

**Table 5. Top DEGs in bovine endometrium at Day 7 post-oestrus (luteal phase) versus oestrus, ranked by ascending B-H FDR.**

| Entrez ID | Gene Name | Gene symbol | Log$_2$ Fold-Change | P-Value | B-H FDR |
|---|---|---|---|---|---|
| **Upregulated** | | | | | |
| 101906669 | uncharacterized LOC101906669 | LOC101906669 | 11.73 | $1.73 \times 10^{-66}$ | $2.93 \times 10^{-62}$ |
| 613835 | mitochondrial ribosomal protein S36 | MRPS36 | 4.51 | $3.61 \times 10^{-52}$ | $3.05 \times 10^{-48}$ |
| 784029 | teratocarcinoma-derived growth factor 1 | TDGF1 | 7.86 | $2.14 \times 10^{-44}$ | $1.21 \times 10^{-40}$ |
| 617104 | family with sequence similarity 162, member A | FAM162A | 3.5 | $3.34 \times 10^{-40}$ | $1.41 \times 10^{-36}$ |
| 104974435 | uncharacterized LOC104974435 | LOC104974435 | 14.27 | $1.82 \times 10^{-38}$ | $6.16 \times 10^{-35}$ |
| 614149 | nudix (nucleoside diphosphate linked moiety X)-type motif 5 | NUDT5 | 2.41 | $1.03 \times 10^{-34}$ | $2.50 \times 10^{-31}$ |
| 287019 | 6-phosphofructo-2-kinase/fructose-2,6-biphosphatase 2 | PFKFB2 | 3.52 | $1.64 \times 10^{-34}$ | $3.46 \times 10^{-31}$ |
| 526200 | absent in melanoma 1 | AIM1 | 3.49 | $7.53 \times 10^{-34}$ | $1.42 \times 10^{-30}$ |
| 100125309 | TSC22 domain family, member 3 | TSC22D3 | 3.06 | $2.90 \times 10^{-33}$ | $4.90 \times 10^{-30}$ |
| 617374 | proteoglycan 3 | PRG3 | 9.49 | $3.83 \times 10^{-33}$ | $5.89 \times 10^{-30}$ |
| **Downregulated** | | | | | |
| 613890 | potassium inwardly-rectifying channel, subfamily J, member 16 | KCNJ16 | -5.77 | $1.40 \times 10^{-36}$ | $3.95 \times 10^{-33}$ |
| 533129 | ets variant 4 | ETV4 | -4.77 | $1.77 \times 10^{-20}$ | $6.80 \times 10^{-18}$ |
| 534731 | EPH receptor B1 | EPHB1 | -3.95 | $2.92 \times 10^{-20}$ | $1.03 \times 10^{-17}$ |
| 506545 | claudin 10 | CLDN10 | -3.44 | $1.46 \times 10^{-18}$ | $3.81 \times 10^{-16}$ |
| 104968606 | laminin subunit alpha-3-like | LOC104968606 | -4.43 | $1.81 \times 10^{-17}$ | $4.01 \times 10^{-15}$ |
| 531699 | aldehyde oxidase 2 | AOX2 | -8.37 | $2.16 \times 10^{-17}$ | $4.68 \times 10^{-15}$ |
| 522184 | potassium inwardly-rectifying channel, subfamily J, member 15 | KCNJ15 | -4.6 | $4.34 \times 10^{-17}$ | $8.75 \times 10^{-15}$ |
| 511211 | sparc/osteonectin, cwcv and kazal-like domains proteoglycan (testican) 1 | SPOCK1 | -3.8 | $6.51 \times 10^{-16}$ | $1.05 \times 10^{-13}$ |
| 100336873 | laminin, alpha 3 | LAMA3 | -3.92 | $1.18 \times 10^{-15}$ | $1.83 \times 10^{-13}$ |
| 508204 | growth regulation by oestrogen in breast cancer 1 | GREB1 | -2.7 | $2.56 \times 10^{-15}$ | $3.76 \times 10^{-13}$ |

**Table 6. Top five diseases and functions categories and canonical pathways based on DEGs from each perioestrous time point versus the onset of oestrus.**

| Time point | Top disease and functions categories | Top canonical pathways |
|---|---|---|
| **12 h post-CIDR removal** | 1 *Cellular movement*<br>2 *Neurological disease*<br>3 *Cellular growth and proliferation*<br>4 *Cancer*<br>5 *Organismal injury and abnormalities* | 1 *Axonal guidance signaling*<br>2 *Superpathway of cholesterol biosynthesis*<br>3 *Superpathway of geranylgeranyldiphosphate biosynthesis I (via mevalonate)*<br>4 *PCP pathway*<br>5 *Colorectal cancer metastasis signaling* |
| **24 h post-CIDR removal** | 1 *Neurological disease*<br>2 *Psychological disorders*<br>3 *Cardiovascular disease*<br>4 *Opthalmic disease*<br>5 *Organismal injury and abnormalities* | 1 *HIF1α signaling*<br>2 *Axonal guidance signaling*<br>3 *Amyotrophic lateral sclerosis signaling*<br>4 *Hepatic fibrosis/hepatic stellate cell activation*<br>5 *Clathrin-mediated endocytosis signaling* |
| **12 h post-oestrus** | 1 *Cellular development*<br>2 *Cellular growth and proliferation*<br>3 *Hematological system development and function*<br>4 *Lymphoid tissue structure and development*<br>5 *Cell-to-cell signaling and interaction* | 1 *Lysine Degradation V*<br>2 *Glycine betaine degradation*<br>3 *IL-12 signaling and production in macrophages*<br>4 *Endoplasmic reticulum stress pathway* |
| **48 h post-oestrus** | 1 *Cancer*<br>2 *Organismal injury and abnormalities*<br>3 *Immunological disease*<br>4 *Connective tissue disorders*<br>5 *Inflammatory disease* | 1 *Superpathway of cholesterol biosynthesis*<br>2 *Cholesterol biosynthesis I*<br>3 *Cholesterol biosynthesis II (via 24,25-dihydrolanosterol)*<br>4 *Cholesterol biosynthesis III (via desmosterol)*<br>5 *LXR/RXR activation* |
| **Day 7 post- oestrus (luteal phase)** | 1 *Cellular growth and proliferation*<br>2 *Cancer*<br>3 *Organismal injury and abnormalities*<br>4 *Cellular movement*<br>5 *Molecular transport* | 1 *Hepatic fibrosis/hepatic stellate cell activation*<br>2 *Axonal guidance signaling*<br>3 *LPS/IL-1 mediated inhibition of RXR function*<br>4 *Fatty acid β-oxidation I*<br>5 *Role of osteoblasts, osteoclasts and chondrocytes in rheumatoid arthritis* |

## IPA diseases and functions categories

A total of 70 IPA categories relating to diseases and functions were enriched at 12 h post-CIDR removal (S4 Table). The top five IPA categories were *Cellular movement*, *Neurological disease*, *Cellular growth and proliferation*, *Cancer*, and *Organismal injury and abnormalities* (Table 6). For the 24 h post-CIDR removal time point versus oestrus, 73 significant IPA categories were identified (S4 Table), with the top five being *Neurological disease*, *Psychological disorders*, *Cardiovascular disease*, *Ophthalmic disease*, and *Organismal injury and abnormalities* (Table 6).

At 12 h post-oestrus, there were 65 significant IPA diseases and functions categories (S4 Table), the most significant of which were *Cellular development*, *Cellular growth and proliferation*, *Hematological system development and function*, *Lymphoid tissue structure and development*, and *Cell-to-cell signaling and interaction*. Seventy-five IPA diseases and functions categories were identified at 48 h post-oestrus (S4 Table), with the top five being *Cancer*, *Organismal injury and abnormalities*, *Immunological disease*, *Connective tissue disorders* and *Inflammatory disease* (Table 6).

Finally, at Day 7 post-oestrus (luteal phase) 70 significant categories of diseases and disorders were identified (S4 Table). Of these, the top five most significant were *Cellular growth and proliferation*, *Cancer*, *Organismal injury and abnormalities*, *Cellular movement*, and *Molecular transport* (Table 6).

## IPA canonical pathways

The most prominent IPA pathways for each time point relative to oestrus are shown in Fig 3. Forty-three canonical pathways were significantly over-represented at 12 h post-CIDR removal (S5 Table). Three of these had a positive z-score, indicating that they displayed

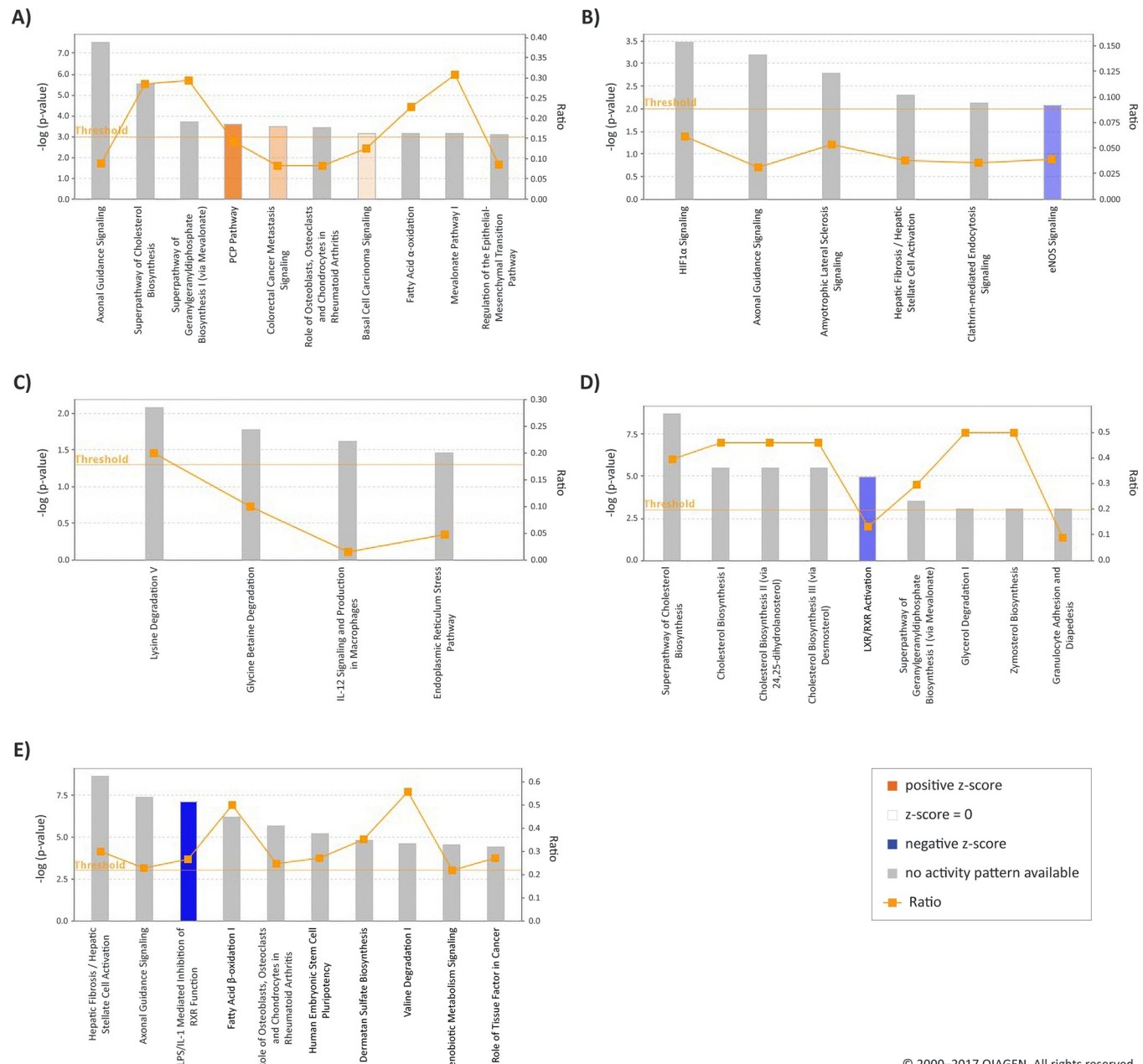

**Fig 3. IPA analysis showing the over-represented canonical pathways at the five time points relative to oestrus.** Pathways associated with significantly modulated genes are shown on the x axis of each bar graph and the -log significance is shown on the y axis. The functions are listed from most significant to less significant (indicated by bar height). Orange bars denote a positive z score, blue bars denote a negative z score. An orange horizontal line denotes the threshold for significance, i.e. p = 0.05. The ratios, which are the number of genes that are modulated in a given pathway divided by the total number of genes in the pathway, are shown on the right-hand side y axis. A) 12 h post-CIDR removal, B) 24 h post-CIDR removal, C) 12 h post-oestrus, D) 48 h post-oestrus, E) Day 7 post-oestrus (luteal phase).

activation (*PCP or planar cell polarity pathway*; *Colorectal cancer metastasis signaling* and *Basal cell carcinoma signaling*). At 24 h post-CIDR removal, 20 over-represented canonical pathways were identified (S5 Table) and the top pathways were *HIF1α signaling, Axonal guidance signaling, Amyotrophic lateral sclerosis signaling, Hepatic fibrosis/hepatic stellate cell activation*, and *Clathrinid-mediated endocytosis signaling*.

At 12 h post-oestrus, only four canonical pathways were over-represented (S5 Table, Fig 3). These were *Lysine degradation V*, *Glycine betaine degradation*, *IL-12 signaling and production in macrophages*, and *Endoplasmic reticulum stress pathway* (Fig 5). A total of 46 over-represented canonical pathways at 48 h post-oestrus (S5 Table) included: *Superpathway of cholesterol biosynthesis*, *Cholesterol biosynthesis I*, *Cholesterol biosynthesis II*, *Cholesterol biosynthesis III*, and *LXR/RXR activation*.

At Day 7 post-oestrus (luteal phase), 124 canonical pathways were significantly over-represented. The top five were *Hepatic fibrosis/hepatic stellate cell activation*, *Axonal guidance signaling*, *LPS/IL-1 mediated inhibition of RXR* function, *Fatty acid β-oxidation I*, and *Role of osteoblasts, osteoclasts and chondrocytes in rheumatoid arthritis*.

### IPA interaction networks

Overall, all perioestrus time points revealed interaction networks involved in similar disease and functions categories such as *Organismal injuries and abnormalities*, *Cancer*, and *Neurological disease* (S6 Table).

At the 12 h post-CIDR time point, 25 interaction networks were identified. The top ranked network had 27 focus molecules, with three genes being upregulated (*ATP12A*, *KSR2*, and *SPINK7*) and the NF-κB complex was at the centre of this network (Fig 4). At 24 h post-CIDR removal, the top interaction network (out of a total of 20) had 26 focus molecules and was also centred on the NF-κB complex (Fig 5).

For the 12 h post-oestrus time point, a total of three interaction networks were identified and the top interaction network, again centred on the NF-κB complex, had 17 focus molecules (Fig 6).

At 48 h post-oestrus there was an increase in significant interaction networks. A total of 25 interaction networks were identified, and the top interaction network had 32 focus molecules (Fig 7). This network was not centred on one gene, but had several hub genes, as did the network for the Day 7 post-oestrus (luteal phase) time point, for which a total of 25 interaction networks were identified, with the top network presenting 35 focus molecules (Fig 8).

### IPA upstream regulators

For the 12 h post-CIDR removal time point, analysis revealed 1,125 upstream regulators (S7 Table). The most highly ranked, by ascending *P*-value, were β-estradiol, TNF, TGF-β1, HDAC (group of histone deacetylases), and the transcription regulator TWIST1 (Table 7). At the 24 h post-CIDR removal time point, the 462 upstream regulators identified (S7 Table) also included HDAC, TGF-β1, and TNF among the top ranked (Table 7).

At 12 h post-oestrus, a total of 107 upstream regulators were identified (S7 Table) that included aldosterone, ESR2 (oestrogen receptor), LHX1 (transcription regulator), TGF-β1, and the drug ciprofibrate as highly ranked (Table 7). At 48 h post-oestrus, the top five upstream regulators were β-oestradiol, the drug dexamethasone, XBP1 (a transcriptional regulator), TGF-β1, and cholesterol (Table 7), with a total of 1,144 upstream regulators (S7 Table). Seven days after oestrus (luteal phase), the number of significant upstream regulators rose to 1,693 (S7 Table) and among the top ranked regulators were β-oestradiol, TGF-β1, dexamethasone, progesterone, and TNF (Table 7).

## Discussion

This study concentrated specifically on the endometrial transcriptome during the perioestrus period and as expected, we observed significant and wide-ranging alterations to the transcriptome. In particular, immune-related changes, as well as alterations that could be re-interpreted

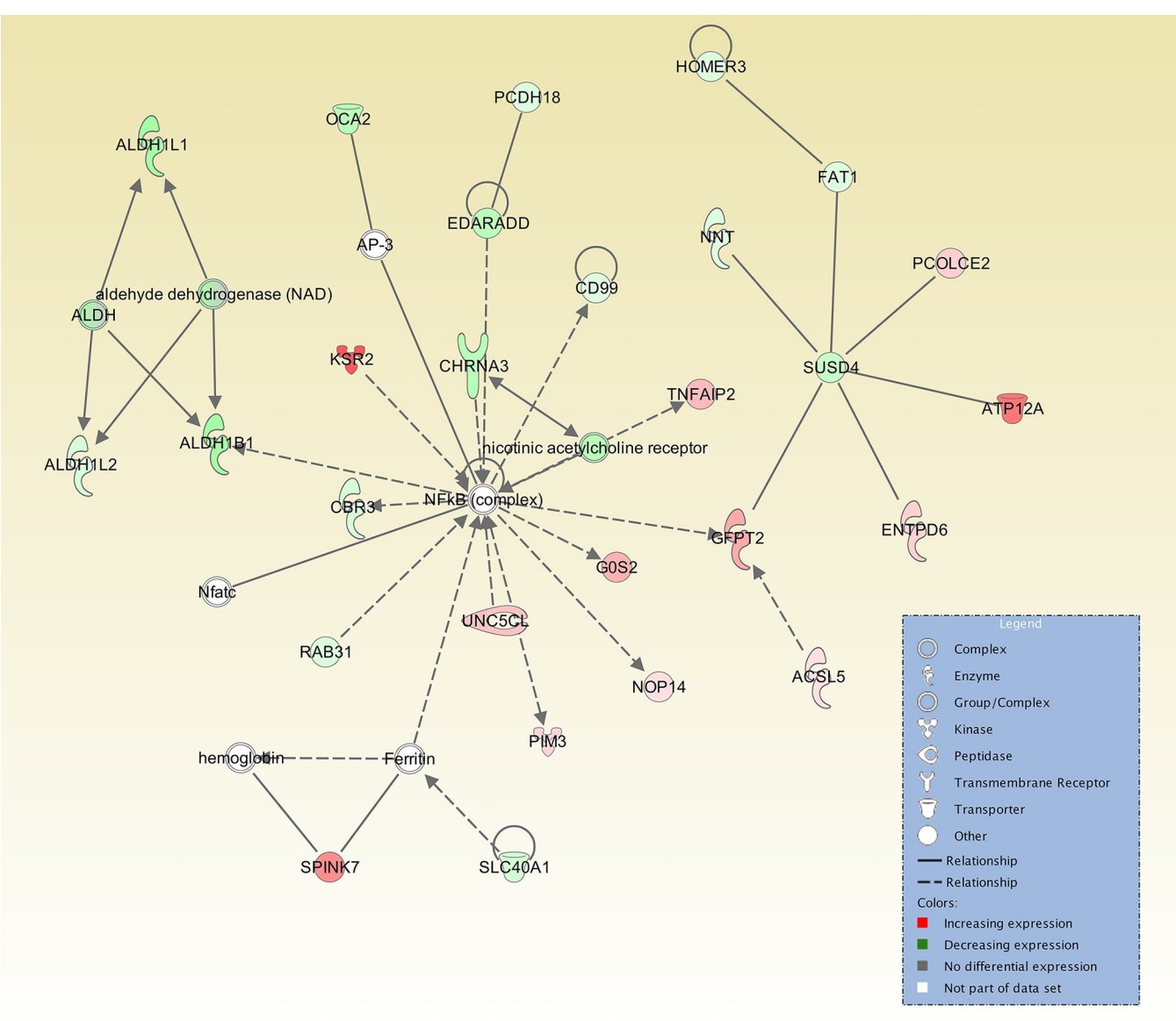

**Fig 4. Top IPA gene interaction network at 12 h post-CIDR removal versus oestrus.** Network 1 is involved in *Endocrine system disorders*, *Hereditary disorders*, and *Organismal injury and abnormalities* (score 34, with 27 focus molecules). The relationship is described as either a direct interaction (solid line) or an indirect interaction (dashed line), whereas the intensity of the colour indicates the level of upregulation (red) or downregulation (green) of the respective molecules. Reprinted from Qiagen IPA under a CC BY license, with permission from Qiagen, original copyright 2017.

as tissue remodelling in this context were evident. The immune-related variations in gene expression would correlate with the time when sperm and seminal fluid would be introduced by mating. Numerous studies of the bovine endometrial transcriptome have focused on fertility, rather than the dynamic changes that occur throughout the cycle. Several studies of the effect of the peri-ovulatory endocrine environment on bovine endometrium have been performed at Days 4 and 7 post-induction of ovulation, which overlaps with the Day 7 time point in this study [17,62,63], but the changes closer to oestrus have undergone less exploration.

Previous studies of the endometrial transcriptome have applied microarray and RNA-seq techniques [3,19,64], some of which focus on differences in fertility [65–67] rather than differences related to precise times in the cycle. In this study, the overall gene expression trend

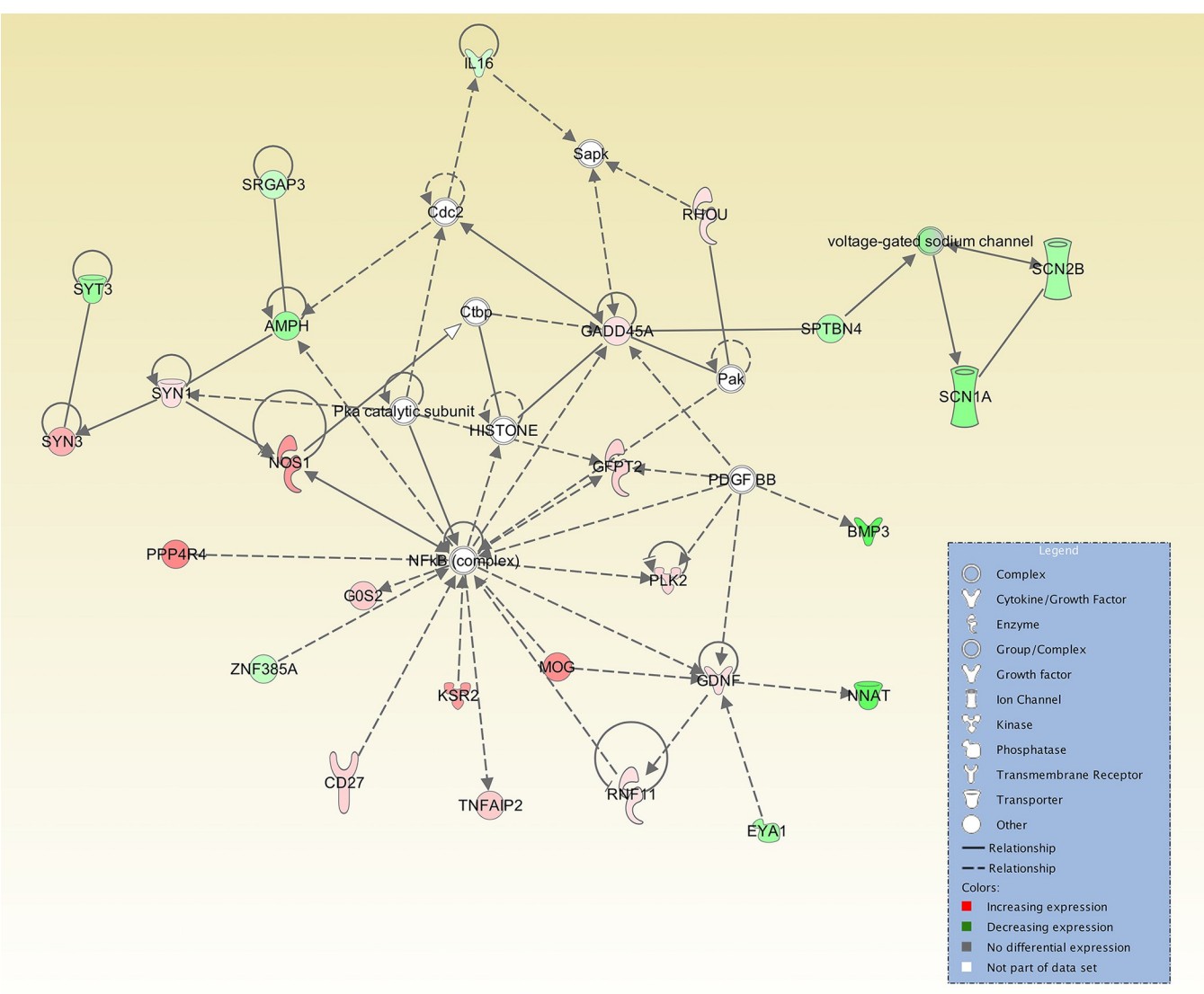

**Fig 5. Top IPA gene interaction network at 24 h post-CIDR removal versus oestrus.** Network 1 is involved in *Neurological disease*, *Nervous System Development and Function and Behaviour* (score 45, with 26 focus molecules). The relationship is described as either a direct interaction (solid line) or an indirect interaction (dashed line), whereas the intensity of the colour indicates the level of upregulation (red) or downregulation (green) of the respective molecules. Reprinted from Qiagen IPA under a CC BY license, with permission from Qiagen, original copyright 2017.

(Fig 2) had parallels with the fluctuations in numbers of DEGs observed in the cervix for the same animals by Pluta et al. [40] and Pluta [68] in which DEGs were highest at Day 7 post-oestrus and lowest for the oestrus +12 h time point. In this study, the lowest number of DEGs (42) was seen at 12 h post-oestrus—when the hormonal environment is very similar to that at oestrus—and the highest number (3,463) was observed at Day 7 post-oestrus (luteal phase). This indicates a significant response to the elevation in P4 between Day 2, when it is very low, similar to that at oestrus [40], and Day 7 of the cycle, when P4 is high. The samples from the cervices of the same animals showed a similar pattern in terms of relative numbers of DEGs, with 339 genes downregulated and 3,798 upregulated at Day 7. In particular, innate and adaptive immunity pathways were upregulated in cervix at the oestrus +48 h time point [68].

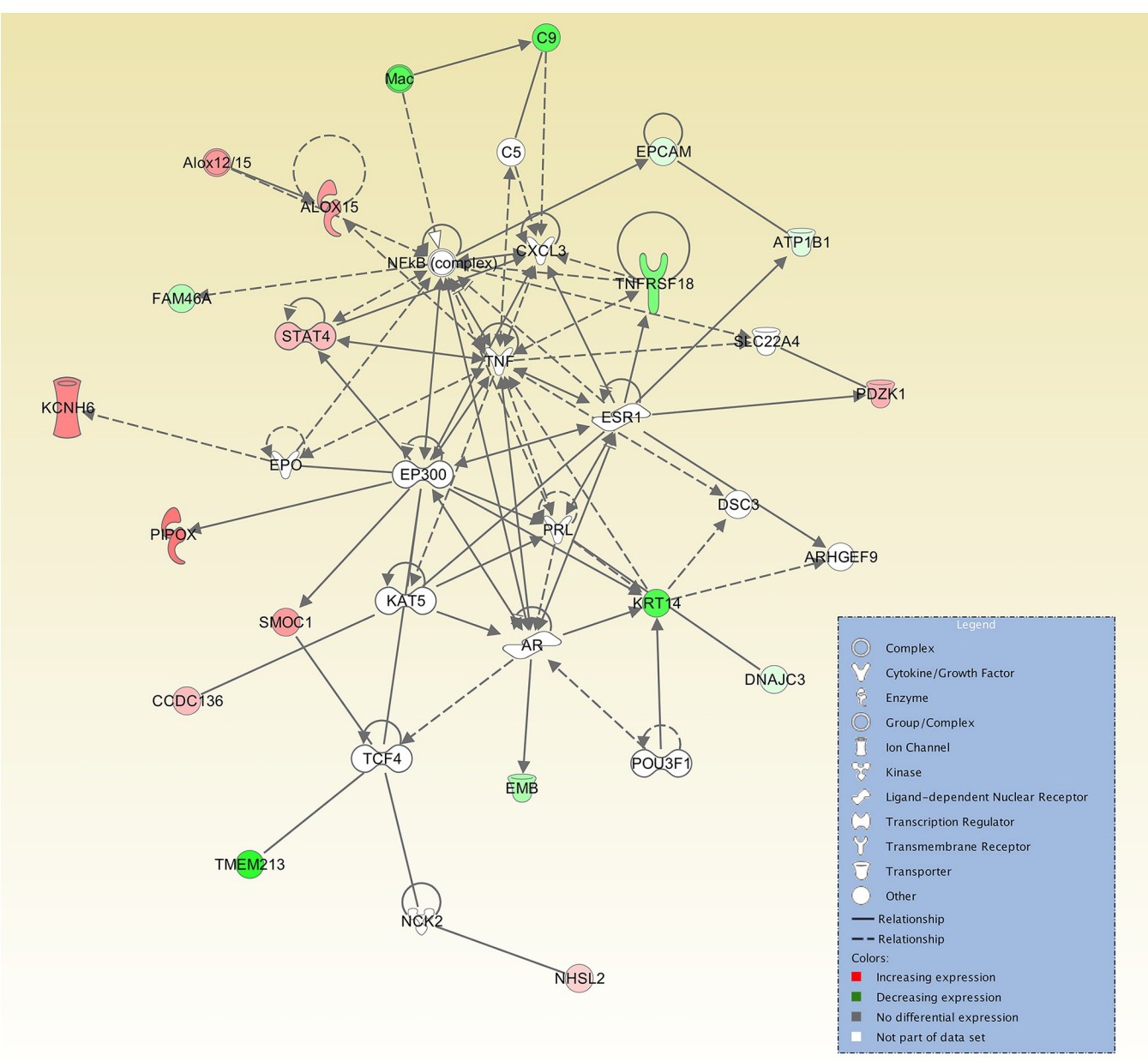

**Fig 6. Top IPA gene interaction network at 12 h post-oestrus versus oestrus.** Network 1 is involved in *Organ Development*, *Reproductive System Development* and *Function and Cell Death and Survival* (score 40, with 17 focus molecules). The relationship is described as either a direct interaction (solid line) or an indirect interaction (dashed line), whereas the intensity of the colour indicates the level of upregulation (red) or downregulation (green) of the respective molecules. Reprinted from Qiagen IPA under a CC BY license, with permission from Qiagen, original copyright 2017.

This study examines the endometrium in its entirety, and not individual cell types. More recent work investigating individual cell types showed that three different cell types isolated from bovine endometrium by laser microdissection at Day 15 of the oestrous cycle demonstrated different molecular signatures for luminal and glandular epithelium, as well as for stromal cells, which showed the greatest differences in gene expression [69]. Differences in these cell types were equally pronounced in post-partum bovine endometrium, and progesterone gave rise to different patterns of expression in the three cell types [70].

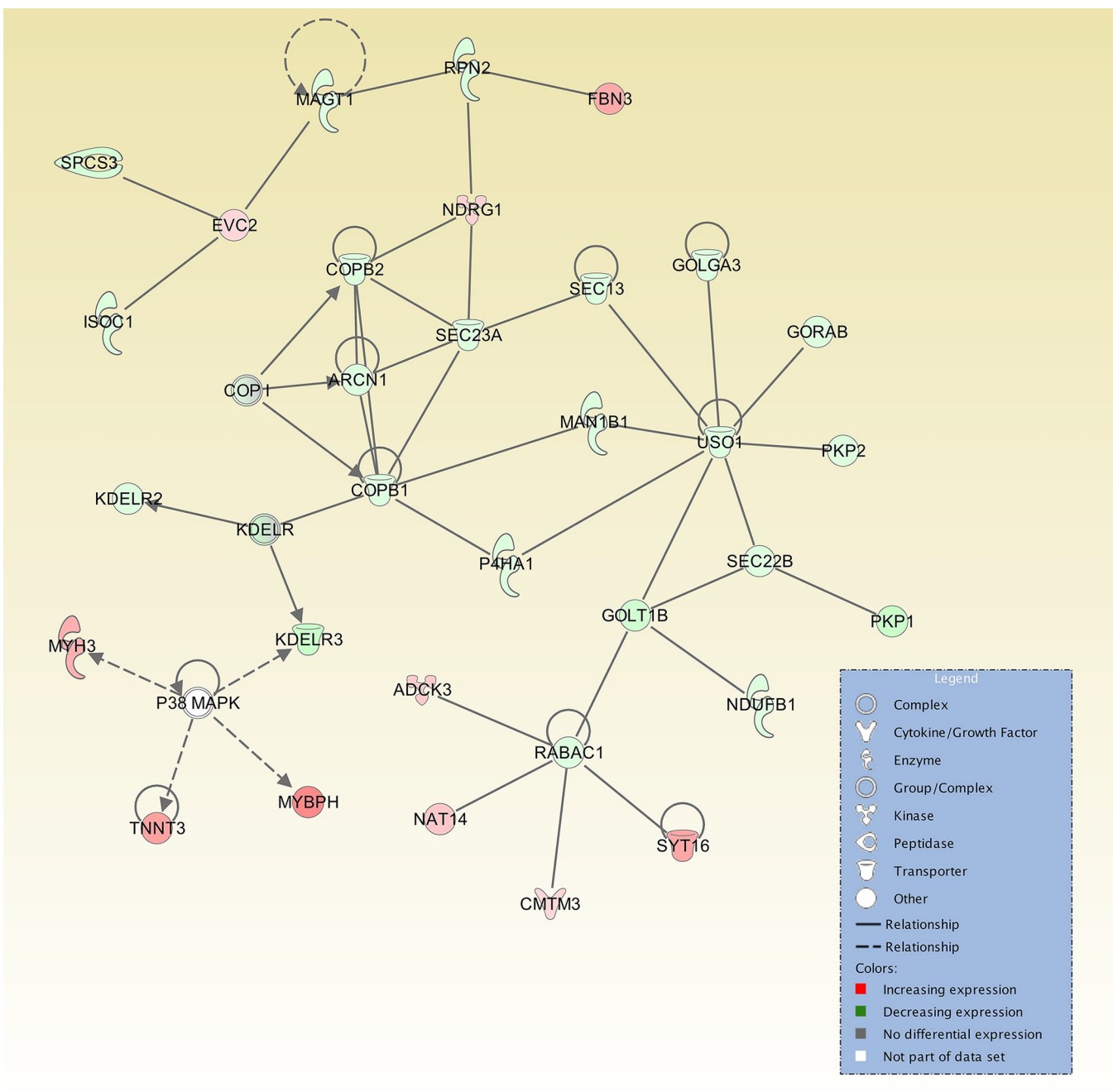

**Fig 7. Top IPA gene interaction network at 48 h post-oestrus versus oestrus.** Network 1 is involved in *Connective Tissue Disorders*, *Developmental Disorder* and *Hereditary Disorder* (score 44, with 32 focus molecules). The relationship is described as either a direct interaction (solid line) or an indirect interaction (dashed line), whereas the intensity of the colour indicates the level of upregulation (red) or downregulation (green) of the respective molecules. Reprinted from Qiagen IPA under a CC BY license, with permission from Qiagen, original copyright 2017.

The variations that we observed were also shown by Bauersachs et al. [64] and Mitko et al. [3] but these showed significant temporal expression variations between Day 0 and Day 12, extending to Day 18 for the latter study, which fell well beyond the time period encompassed by this study. Forde et al. [31] identified 3,969 DEGs between Days 7 and 13, whereas the

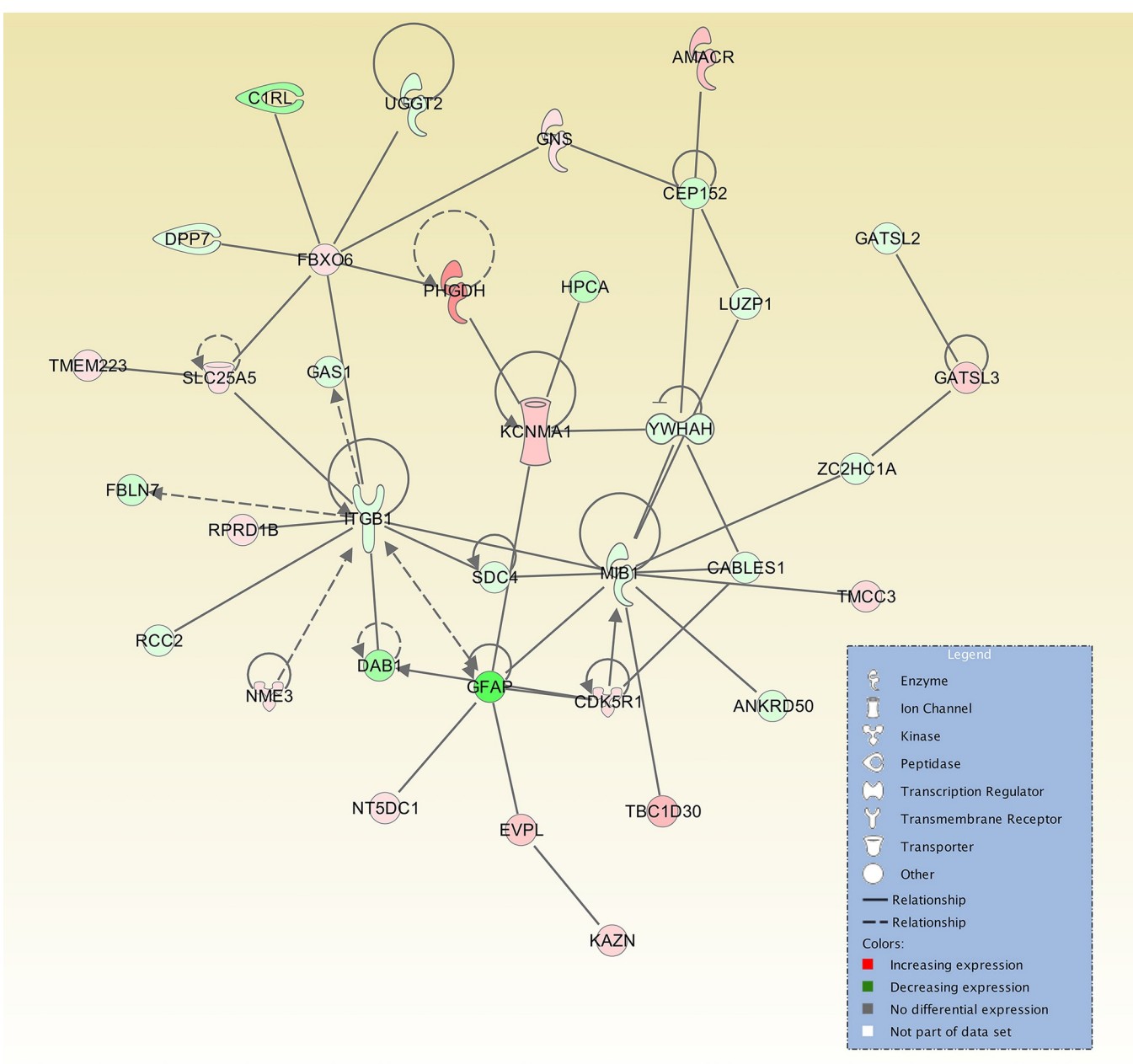

**Fig 8. Top IPA gene interaction network at Day 7 post-oestrus (luteal phase) versus oestrus.** Network 1 is involved in *Cellular Assembly and Organization*, *Developmental Disorder*, *Nervous System Development and Function* (score 32, with 35 focus molecules). The relationship is described as either a direct interaction (solid line) or an indirect interaction (dashed line), whereas the intensity of the colour indicates the level of upregulation (red) or downregulation (green) of the respective molecules. Reprinted from Qiagen IPA under a CC BY license, with permission from Qiagen, original copyright 2017.

comparison between Days 5 and 7 yielded only seven DEGs, implying little change in gene expression over that time. While all these studies, along with the present study, all illustrate clear differences in expression that are dependent on time and on hormonal environment, there have been fewer studies of the precise nature of these differences immediately preceding and post-oestrus.

**Table 7. The five top IPA upstream regulators, based on DEGs from each of the perioestrous time points versus the onset of oestrus.**

| Time point | Top upstream regulators | Molecule type | Predicted activation state | P-value |
|---|---|---|---|---|
| **12 h post-CIDR removal** | 1 ß-estradiol<br>2 TNF<br>3 TGF-ß1<br>4 HDAC<br>5 TWIST1 | 1 Chemical endogenous<br>2 Cytokine<br>3 Growth factor<br>4 Group of histone deacetylases<br>5 Transcription regulator | 1 Activated<br>2 Activated<br>3 Neutral<br>4 Neutral<br>5 Neutral | $6.41 \times 10^{-21}$<br>$9.74 \times 10^{-15}$<br>$5.99 \times 10^{-12}$<br>$8.33 \times 10^{-12}$<br>$2.79 \times 10^{-10}$ |
| **24 h post-CIDR removal** | 1 HDAC<br>2 TGF-ß1<br>3 TNF<br>4 Genistein<br>5 Thioacetamide | 1 Group of histone deacetylases<br>2 Growth factor<br>3 Cytokine<br>4 Chemical drug<br>5 Chemical toxicant | 1 Neutral<br>2 Neutral<br>3 Neutral<br>4 Neutral<br>5 Neutral | $6.08 \times 10^{-08}$<br>$1.26 \times 10^{-07}$<br>$3.02 \times 10^{-06}$<br>$4.14 \times 10^{-06}$<br>$1.19 \times 10^{-05}$ |
| **12 h post-oestrus** | 1 Aldosterone<br>2 ESR2<br>3 LHX1<br>4 TGF-ß1<br>5 Ciprofibrate | 1 Chemical endogenous<br>2 Ligand-dependent nuclear receptor<br>3 Transcription regulator<br>4 Growth factor<br>5 Chemical drug | 1 Neutral<br>2 Neutral<br>3 Neutral<br>4 Neutral<br>5 Neutral | $3.88 \times 10^{-05}$<br>$4.35 \times 10^{-04}$<br>$5.60 \times 10^{-04}$<br>$6.71 \times 10^{-04}$<br>$1.49 \times 10^{-03}$ |
| **48 h post-oestrus** | 1 ß-estradiol<br>2 Dexamethasone<br>3 XBP1<br>4 TGF-ß1<br>5 Cholesterol | 1 Chemical endogenous<br>2 Chemical drug<br>3 Transcription regulator<br>4 Growth factor<br>5 Chemical endogenous | 1 Inhibited<br>2 Neutral<br>3 Inhibited<br>4 Neutral<br>5 Activated | $3.53 \times 10^{-24}$<br>$3.93 \times 10^{-22}$<br>$3.43 \times 10^{-16}$<br>$2.79 \times 10^{-14}$<br>$1.87 \times 10^{-13}$ |
| **Day 7 post- oestrus (luteal phase)** | 1 ß-estradiol<br>2 TGF-ß1<br>3 Dexamethasone<br>4 Progesterone<br>5 TNF | 1 Chemical endogenous<br>2 Growth factor<br>3 Chemical drug<br>4 Chemical endogenous<br>5 Cytokine | 1 Inhibited<br>2 Inhibited<br>3 Neutral<br>4 Neutral<br>5 Inhibited | $1.42 \times 10^{-34}$<br>$2.40 \times 10^{-32}$<br>$1.52 \times 10^{-31}$<br>$1.38 \times 10^{-30}$<br>$3.13 \times 10^{-28}$ |

Molecule types, predicted activation state and *p*-values denoting significance are shown for each upstream regulator.

This work showed significant differences in the overall pre- and post-oestrus patterns of expression of both individual genes and at a pathway and regulatory level. At a gene level, three genes were differentially expressed across each of the five perioestrus time points (S3 Table): collagen type VI alpha 6 (*COL6A6*), mucin-6 (*LOC101903030*), and angiotensinogen (*AGT*). *COL6A6* was upregulated at all time points, displaying the largest $\log_2$ fold-change at Day 7 post-oestrus (luteal phase) (+9.05). Collagens are diverse and are involved in organizing the extracellular matrix in the endometrium of pregnant bovines, as well as participating in signalling and adhesion [46]. Collagen VI forms a subepithelial and pericellular fibril sheet in both superficial and deep bovine endometrium, linking cells to the collagen fibril network and localizing around capillaries and smooth muscle cells [71]: therefore, in this case it could have a function in reorganizing cells, particularly the vasculature of the endometrium during this period. However, its expression is low ($\log_2$ CPM 0.0597 at Day 7) so variations in expression of other collagen genes may be of more biological relevance.

The mucin 6, oligomeric mucus/gel-forming gene (*MUC6*, *LOC101903030*) encodes a secreted gel-forming mucin, and its expression was downregulated across almost all perioestrus time points, except for Day 7 post-oestrus (luteal phase), where it was upregulated (S3 Table). This gene was not expressed in cervical tissues from the same animals at any time point [40]. A similar trend in *MUC6* expression was observed in human endometrium, with lower expression in proliferative endometrium, but not in menstrual or secretory endometrium [72]. *MUC6* is strongly expressed in the human stomach and pancreas and appears to have a protective effect against acid and proteolytic digestion in stomach [73]. In bovine endometrium it may have a protective function after oestrus, when the endometrial thickness is relatively low compared to that before oestrus [74].

In contrast, the angiotensinogen gene (*AGT*) was upregulated at all perioestrus time points, with the largest log$_2$-fold increase relative to oestrus seen at Day 7 post-oestrus (+9.43) (S3 Table). This gene encodes pre-angiotensinogen, a component of the renin-angiotensin system, cleaved by renin to angiotensin I and further cleaved to the active enzyme angiotensin II. which promotes cell proliferation, angiogenesis, fibrosis, migration, and invasion [75]. Shimizu et al. [35] demonstrated that *AGT* was part of a gene cluster downregulated by E2 and upregulated by P4, in accordance with elevated expression at Day 7 in this study. An analysis of data from the cervices of the same animals also showed differences in *AGT* expression, which was elevated in the luteal phase relative to the follicular phase [42]. In human endometrial stromal cells, decidualization was associated with an increase in expression of prorenin and other components of the renin-angiotensin system [76], suggesting that *AGT* upregulation is essential to facilitate decidualization. AGT increases nitric oxide production in endothelial cells of the ovine fetoplacental artery, implying a role in endothelial development and vasoconstriction [77].

## Tissue remodelling and innervation in the proliferative phase

The thickness of the bovine endometrium increases during the follicular phase in an induced oestrus cycle [78]. In addition, the uterine wall thickness was shown to increase as the corpus luteum regressed, and decreased after ovulation [79], accompanied by more frequent cell divisions in the epithelium and stroma during the proliferative phase, with more ciliated epithelial cells and endometrial gland branching [15]. The changes in gene expression observed in the pre-oestrus timepoints of the present study support these changes.

For example, some of the top disease and functions categories, and canonical pathways at 12h post-CIDR removal (Table 6) facilitate endometrial cell proliferation. Proliferating cells move to enable increase in cell numbers and thickness of the endometrium [74], and we hypothesize that this is represented by the *Cellular movement*, *Cellular growth and proliferation*, *Cellular Development* and *Tissue Development* categories (S4 Table). Common to these four categories are the genes *DPYSL2* and *DBN1*, while *NTN1* appears in the first two. *NKD1*, *COL14A1*, and *ADAMTS8* also appear frequently in the 12 top ranked IPA categories. *DPYSL2* appears in 14 of the top 25 categories. *DBN1* encodes an actin-binding protein involved in neural growth, and *NKD1* encodes a protein that negatively regulates the Wnt/β-catenin signalling pathway [80] which is hormonally regulated in the uterus [81]. In terms of canonical pathways, the activation of the *Planar cell polarity pathway* (PCP, Fig 3A) is key to establishing cellular asymmetry and a functional barrier regulating intercellular adhesion, signalling, and a functional cytoskeleton, and in human endometrium, the planar polarity is lost in luminal epithelium by the mid-secretory phase of the menstrual cycle, which allows for receptivity [82] and which may be its role in this case.

Changes in innervation may be reflected by the appearance of *DPYSL2* and *NTN1*, which were both part of the *Axonal guidance signaling* canonical pathway, ranked number 1 for CIDR+12h and number 2 for CIDR+24h. (S5 Table). *DPYSL2* encodes the dihydropyrimidinase like 2 protein—a member of the collapsin response mediator protein family—which interacts with calcium channels and is required for normal axonal outgrowth [83]. *NTN1* encodes netrin-1—an axonal guidance regulatory protein—which guides axons to their final destination [84], and in human endometrium, netrin-1 may be involved in changing endometrial gland architecture from a proliferating to a secretory phase [85]. Oestradiol can affect neuronal function in different ways, for example by stimulating neurite growth, modulating the expression of neurotransmitters and controlling the survival of neurons [86], but in many mammalian species, the endometrium is not well innervated: for example, in human endometrium, nerve fibres are only present in the basal layer [87].

It is also noteworthy in this context that the *ST8SIA2* gene is upregulated at the time points prior to oestrus. This gene encodes a sialyltransferase that makes polysialic acid, which is associated with several proteins, including the neural cell adhesion molecule (NCAM), as well as with a range of immune cells [88]. All these changes are consistent with alterations in innervation of the endometrium and changes that facilitate contractility.

At 24 h post-CIDR removal (Table 6) the *Neurological disease* and *Organismal injury and abnormalities* categories were also enriched, and the *eNOS signaling* pathway was downregulated at 24 h post-CIDR removal (S5 Table). In female horses, there is a dose-dependent inhibitory effect of nitric oxide (NO) on uterine contractility, and it was proposed that NO could diffuse into the myometrium and decrease contractility [89]. The most highly ranked canonical pathway at the CIDR+24 h time point was *HIF1α signaling* (Table 6). HIF1α is the main regulator of adaptation to hypoxia, upregulating genes essential for cell survival and vascularization. In humans, it is needed for repair of the endometrial surface after menstruation occurs [90]. In our dataset, *NOS1*, *MMP14*, *VEGFB*, *MMP11*, *PIK3R2*, *ELOC*, and *PGF* are all associated with this pathway (S5 Table). Matrix metalloproteases are involved in extracellular matrix breakdown, while *VEGFB* encodes a vascular endothelial growth factor. These processes are all essential for endometrial remodelling.

The *Hepatic fibrosis/hepatic stellate cell activation* pathway is listed as one of the five top canonical pathways for the CIDR +24 h time point. This was also observed to be activated by [91] in their study of follicular stage endometrium. This pathway in liver leads to wound healing and deposition of extracellular matrix components: it is also driven by TGF-β signalling [92] and in our study included the genes *IGF1*, *EDNRB*, *VEGFB*, *MYL6B*, *PDGFD*, *COL6A6*, and *PGF*.

For both time points before oestrus, transforming growth factor beta 1 (TGF-β1), the cytokine tumour necrosis factor (TNF), and HDAC (group of histone deacetylases) were among the top ranked upstream regulators (Table 7). TGF-β1 is required for the development of smooth muscle in the female reproductive tract [93], as well as being present in seminal fluid and altering the endometrium and oviduct to promote embryonic development and implantation [94]. In humans, expression of TNF has also been shown to rise in the mid- to late-proliferative phases [95], while in bovine endometrial cells, TNF-induced production of chemokines [96], directs the migration of leukocytes to inflammation sites [97]. This observation is consistent with the upregulation of genes associated with immunity seen at the pre-oestrus timepoints. These genes include *CLEC10A*, a gene encoding a C-type lectin domain-containing protein that is a marker for human CD1c+ dendritic cells that enhance TLR 7/8 – induced cytokine secretion [98] and *SLAMF7*, which encodes a receptor that regulates the activation of natural killer cells in humans [99]. These observations suggest that endometrial immunity is upregulated at this timepoint, actively preparing for sperm deposition and pathogen entry. The top interaction networks for CIDR + 12 h and CIDR +24 h are centred on the transcription factor NFκB complex (Figs 4 and 5) although these are different networks. For the CIDR+12 h top network, involved *in Endocrine system disorders*, *Hereditary disorders and Organismal injury and abnormalities*, NF-κB is linked to various hub genes, such as *SUSD4*, which negatively regulates complement activation, and *ALDH1B1*, which links to various aldehyde dehydrogenases. The top network for CIDR+24 h is *Neurological disease*, *Nervous System Development and Function and Behavior*, with links to voltage-dependent $Na^+$ channels and *SYN* genes, encoding synapsins that are associated with the trafficking of synaptic vesicles.

Histone deacetylases, listed as upstream regulators at the CIDR +12 h and CIDR +24 h time points (Table 7) remove acetyl groups from lysine residues of histones, thereby making DNA less accessible to transcription factors and repressing transcription of genes. Suppression of these enzymes can induce apoptosis, cell cycle arrest, and differentiation. *TWIST1* is listed as a

transcriptional regulator at CIDR +12 h: this gene is associated with cancer metabolism and regulates the epithelial-to-mesenchymal transition [100]. The role of these regulators is consistent with the top diseases and functions categories that include *Cancer*, *Cellular growth and proliferation* and *Organismal injury and abnormalities*, supporting the cellular growth essential to the increased endometrial thickening observed in the bovine follicular phase in both natural and induced oestrus [74].

## Changes occurring post-oestrus

Gene expression at the 12 h post-oestrus time point shows little variation compared to that at oestrus, with a total of 42 DEGs between the time points. This is similar in terms of relative numbers of DEGs to observations of cervical tissue, in which only 28 transcripts were downregulated and 259 upregulated compared to the remaining groups [68]. At 12 h post-oestrus only four IPA canonical pathways appear in the dataset: *Lysine degradation V*, *Glycine Betaine degradation*, *IL-12 Signaling and production in macrophages*, and *Endoplasmic reticulum stress pathway*. The gene *PIPOX* is associated with both the *Lysine Degradation V* and *Glycine Betaine degradation* pathways, encoding pipecolic acid oxidase. This gene is upregulated in repeat breeder cows, compared to non-repeat breeder cows during the mid-luteal phase [65], is involved in the metabolism and degradation of sarcosine and L-pipecolic acid and has been hypothesized to protect cells from oxidative stress [101], which may also be its function in endometrium, promoting cell survival. In this environment, activated neutrophils could potentially damage tissues with release of reactive oxygen species. This protection would be in line with the hypothesis that the endometrium undergoes transcriptional alterations to prepare for sperm movement through the uterus towards the oviducts after mating and to protect it from entry of pathogens.

The *Endoplasmic reticulum stress pathway* is also present among the IPA canonical pathways highlighted at oestrus +12h, with downregulation of some of the genes in the pathway (S4 Table). The ER stress pathway is inhibited by E2 in human endometrial cells [102], and as E2 levels are high, comparable to oestrus, this would be consistent with the results in our dataset. Studies of human endometrium [103] showed that E2 can regulate the levels of the heat shock protein GRP78, a component of the UPR, that is suppressed during the E2-dominated phases of the cycle, which may contribute to angiogenesis, cell proliferation, apoptosis, and protein secretion [103].

## Inflammatory response post-oestrus

Inflammation is an essential component of the oestrous cycle. When sperm arrive in the uterus after mating, the majority of the sperm cells are removed through phagocytosis by polymorphonuclear leukocytes, while these cells remove the debris and bacteria introduced by mating [12]. The introduction of seminal plasma upregulates inflammatory mediators in the bovine endometrium [104] indicating that this is also an essential process required to protect the uterus and promote fertility. Male fertility can affect this response, with insemination by high-fertility bulls causing alterations in the transcriptomic response relative to these induced by low-fertility bulls, particularly with respect to immune system genes [37].

However, the present study reveals mediation of the inflammatory response in the absence of sperm or seminal plasma through the enrichment of the *IL-12 signaling and production in macrophages* pathway at 12 h post-oestrus (Table 6). This pathway is involved in the upregulation of STAT4, interferon-gamma production, and a Th1 response that kills bacteria [105]. Our data showed that *STAT4* expression was upregulated at 12 h post-oestrus (Table 3), and it appears consistently among the genes associated with numerous IPA diseases and functions

categories (S4 Table). Among the top DEGs at oestrus +12 h was *ALOX15*. This gene encodes arachidonate 12/15 lipoxygenase, constitutively expressed in uterus, which not only metabolizes arachidonic acid, leading to synthesis of products that can regulate and enhance angiogenesis [106] but affects macrophages, promoting synthesis of phospholipid oxidation products that help to remove apoptotic cells, and lipid precursors required to make pro-resolving mediators involved in inflammation resolution [107]. Another top DEG at the oestrus +12 h time point was *LOC100297044*, predicted to encode *CXCL14*, a cytokine which *in vitro* stimulates uterine natural killer (NK) cells. Administration of this cytokine led to a chemoattractive effect that caused NK cells to cluster around uterine epithelial glands [108]. This cytokine was hypothesized to be important in protecting the host from pathogens, as well as playing a role in decidualization and reconstruction of blood vessels [109] and would thus be consistent with increased immune activity at this time point.

## Lipid metabolism

Lipid metabolism is an important theme that emerges from our dataset. At the CIDR +12h time point, the *Superpathway of cholesterol biosynthesis* and *Superpathway of geranylgeranyl diphosphate biosynthesis I (via mevalonate)* canonical pathways are in an activated state (Tables 6 and S5). Production of cholesterol at this stage would facilitate formation of cell membranes as well as hormone biosynthesis. Pathways relating to immune surveillance and lipid metabolism dominate the data at 48 h post-oestrus relative to oestrus, with the appearance of *Immunological disease* and *Inflammatory disease* among the top five IPA diseases and functions categories (Table 6), while the top ten include *Inflammatory response* and *Immune Cell Trafficking* (S4 Table). This suggests that even in the absence of sperm, endometrial immunity is boosted to provide an environment that defends robustly against introduction of sperm and microorganisms in the period directly after mating. This is consistent with the observation that in cervical samples from the same animals, multiple innate and adaptive immunity pathways are activated at 48 h post-oestrus, including the NK pathway, NOD and TOLL-related innate immune effector pathways [68] suggesting a coordinated response along the reproductive tract. The upregulation of immune-related genes at this time point (Table 4), such as *CD68*, encoding a scavenger receptor found on macrophages, and *SLAMF7*, which is involved in regulating and connecting elements of the innate and adaptive immune responses, activating cytotoxic cells and suppressing inflammation in sepsis [110] is in line with the pathway analysis.

At the same time, there is strong representation of pathways related to cholesterol biosynthesis in the IPA top canonical pathways. Four of the top five belong in this category, with the fifth being *LXR/RXR activation*. A significant proportion of the genes in these pathways are downregulated (39% in the *Superpathway of Cholesterol Biosynthesis*, shown in S4 Table) and among the top 10 downregulated DEGs in Table 4 is *HMGCS1* (3-hydroxymethylglutaryl-CoA synthase). Liver X receptors (RXR) play an important role in controlling lipid metabolism at a transcriptional level, modulating cholesterol and fatty acid metabolism, and activating cholesterol sensors [111]. They also modulate immune responses in macrophages. Agonists of this receptor promote cholesterol efflux and inhibit inflammatory processes *in vivo* [112]. LXR activation promotes fatty acid biosynthesis through pathways that lead to activation of fatty acid synthase (FASN) and stearoyl-coenzyme A desaturase (SCD1) [111]. The top IPA canonical pathway of *LXR/RXR activation* is clearly downregulated (Fig 3), implying that downregulation of lipid metabolism and upregulation of inflammatory processes occurs at this point.

In our data, the IPA analysis demonstrated that cholesterol was among the top five upstream regulators ranked at 48 h post-oestrus and was in an activated state, in contrast to

the other four identified at that time point (Table 7). Other enriched upstream regulators included SREBF2 and SCAP (S7 Table), which form a complex in the endoplasmic reticulum membrane and control cholesterol homeostasis [113].

In our study, there was downregulation of *HMGCS1*, which encodes an enzyme essential for cholesterol synthesis, and *SCD*, which encodes stearoyl CoA desaturase, an endoplasmic reticulum enzyme that catalyses the rate-limiting step in formation of monounsaturated fatty acids. In normal human endometrium this enzyme is highly expressed in the proliferative phase; its knockdown induced apoptosis and decreased the growth of endometrial cancer cells [114].

In addition, studies of the endometrial transcriptome from other species support the idea that lipid metabolism and immune signalling are important at this time. For example, Kim et al. [115] studied the porcine ovary, endometrium, and oviduct transcriptomes throughout the oestrous cycle and found that KEGG pathways relating to immune signalling, biological oxidation, and acute inflammatory response were enriched, indicating that upregulation of these pathways may occur across species, not just cattle.

Lipid synthesis pathways were shown to be essential to provide lipids in uterine histotroph during lactation, as well as regulating the biology of the conceptus at the time of elongation [116]. A recent machine-learning study of multiple transcriptome datasets seven days post-oestrus comparing receptive and non-receptive animals showed down-regulation of lipid modification and fatty acid oxidation [117]. This study also showed that cell cycle processes were downregulated, whereas apoptosis, regulation of apoptosis and Wnt receptor signalling were upregulated [117].

## Day 7 post-oestrus (luteal phase)

The transition to dioestrus (represented in our data by the Day 7 post-oestrus relative to the oestrus time point) is characterised by a major change in hormonal environment and in endometrial morphology. The area represented by gland ducts increases [15], consistent with a secretory state, and longer microvilli and secretory droplets appear in the apical cytoplasm [118]. In our data, the IPA diseases and functions categories included *Cancer* and *Organismal injury and abnormalities*, as seen at the 48 h post-oestrus timepoint. However, *Cellular movement* and *Molecular transport* now appear among these categories, consistent with the transition to a secretory state.

The highest ranked IPA diseases and functions at this time point relative to oestrus are *Cellular movement*, *Neurological Disease*, and *Cellular Growth and Proliferation* (Table 6). These categories include the genes *DPYSL2*, and *DBN1*, which encodes a cytoplasmic actin-binding protein involved in neural growth, which is elevated in human endometrium under high P4 [119]. The top IPA interaction network at Day 7 post-oestrus is involved in *Cellular Assembly and Organization*, *Developmental Disorder*, *Nervous system Development and Function* (Fig 8), consistent with these findings.

The observed changes are reflected in the top IPA over-represented canonical pathways, which include *Hepatic Fibrosis/Hepatic Stellate Cell Activation* and *Axonal Guidance Signaling*. The next over-represented canonical pathway is clearly downregulated (Fig 3) and is *LPS/IL-1 Mediated Inhibition of RXR Function*. This is illustrative of downregulation of the immune system at this time in the cycle. LXR/RXR regulate lipid synthesis and transport, induce genes involved in cholesterol transport, and modulate immune and inflammatory responses in macrophages, inhibiting inflammation *in vivo* [112]. LPS and IL-1 are pro-inflammatory, inhibiting RXR, so the opposite effect is seen to that at the 48h post-oestrus time point.

At the level of individual DEGs (Table 5) the upregulation of *TSC22D3*, which encodes an immunosuppressive transcriptional regulator [120], is also consistent with inhibition of

inflammation. The changes in endometrial structure are illustrated clearly by the downregulated DEGs. These include *CLDN10*, encoding a tight junction component, *LAMA3*, encoding alpha-laminin 3, which regulates cell growth and motility and acts as an adhesin, connecting two layers of basement membrane, and *SPOCK1*, involved in cell-cell and cell-matrix interactions. The highest ranked DEGs in terms of log$_2$ fold change are *LOC101906669* and *LOC104974435*, which are currently uncharacterized bioinformatically predicted gene loci in the NCBI Gene database.

Along with this, canonical pathways including *Role of osteoblasts*, *osteoclasts and chondrocytes in rheumatoid arthritis* and *Fatty acid β-oxidation I* were enriched at Day 7 post-oestrus (Table 6). IPA analysis demonstrated that the second most highly ranked network at Day 7 post-oestrus was involved with *Amino acid metabolism* (S6 Table). The concentrations of amino acids in the uterine lumen vary with day of the oestrus cycle, with glycine and taurine being most abundant, and the availability of amino acids in the uterine lumen appears to be essential to allow an embryo to survive and develop [121]. Glycine may help to regulate the pH, and this study suggested that taurine may be needed for development of the embryo from morula to blastocyst. Furthermore, the concentration of valine in uterine fluid was also positively correlated with increased P4 [122]. The upregulation of small molecule biochemical pathways was consistent with the observations of [91]. França et al. [123] also demonstrated endocrine effects on endometrial amino acid metabolism. They compared beef cattle with high and low pre-ovulatory follicles at Days 4 and 7 of the oestrus cycle and showed that animals producing higher levels of P4 had upregulated solute transporters, whereas the animals with lower P4 had lower concentrations of valine and cystathionine in uterine fluid. Tríbulo et al. [124] characterized the metabolome of uterine fluid between oestrus and Day 7, demonstrating that peak intensity for all metabolites in uterine fluid varied over this time, with the highest concentrations of amino acids at Day 7. The most abundant of these were tryptophan, tyrosine, leucine, phenylalanine, and aspartic acid. There were also large differences in the metabolome between Days 0 and 7, with intermediate values seen at Days 3 and 5. These observations are in agreement with the data in the present study.

In conclusion, this study supported previous work indicating significant hormone-driven changes in the bovine endometrium, but also demonstrated significant variations over a narrow range of time points close to oestrus. In the 48 hours after oestrus these alterations in the transcriptome assist in preparing the endometrium for an influx of sperm and associated pathogens, even in the absence of insemination. We also demonstrate an important role for expression of genes relevant to innervation in the follicular phase and underline the importance of lipid metabolism pathways both before and after oestrus. One caveat of our study is that it examines the overall endometrium and not individual cell types: a future study limited to perioestrus time points would provide more detail if the three main endometrial cell types were disaggregated and used for single-cell RNA-seq analysis [125].

## Supporting information

**S1 Table. Endometrial tissue sample information with total RNA sample quality and quantity.**
(XLSX)

**S2 Table. A. RNA-seq alignment and B. gene assignment statistics.**
(XLSX)

**S3 Table. List of all significant DEGs at each time point relevant to oestrus.** Differentially expressed (DE) genes, ranked by ascending FDR, at A. 12 h post-CIDR, B. 24 h post-CIDR, C.

12 h post-oestrus, D. 48 h post-oestrus, and E. Day 7 post-oestrus (luteal phase) relative to the onset of oestrus are shown.
(XLSX)

**S4 Table. Full list of IPA diseases and functions categories.** IPA disease and functions categories at A. 12 h post-CIDR, B. 24 h post-CIDR, C. 12 h post-oestrus, D. 48 h post-oestrus, and E. Day 7 post-oestrus (luteal phase) are shown relative to the onset of oestrus.
(XLSX)

**S5 Table. Full list of IPA canonical pathways at all time points relative to oestrus.** IPA canonical pathways at A. 12 h post-CIDR, B. 24 h post-CIDR, C. 12 h post-oestrus, D. 48 h post-oestrus, and E. Day 7 post-oestrus (luteal phase) are shown relative to the onset of oestrus.
(XLSX)

**S6 Table. Full list of IPA interaction networks at all time points relative to oestrus.** IPA interaction networks at A. 12 h post-CIDR, B. 24 h post-CIDR, C. 12 h post-oestrus, D. 48 h post-oestrus, and E. Day 7 post-oestrus (luteal phase) are shown relative to the onset of oestrus.
(XLSX)

**S7 Table. Full list of IPA upstream regulators at all time points relative to oestrus.** IPA upstream regulators at A. 12 h post-CIDR, B. 24 h post-CIDR, C. 12 h post-oestrus, D. 48 h post-oestrus, and E. Day 7 post-oestrus (luteal phase) are shown relative to the onset of oestrus. Regulators with predicted activated state are shaded in orange, those with predicted inhibited state are shaded in blue, and those with a predicted neutral state are not shaded.
(XLSX)

## Acknowledgments

The authors would like to thank Gillian Mc Hugo for her assistance with IPA analysis, Professor Patrick Lonergan, UCD School of Agriculture and Food Science, and Dr Marion Ryan, UCD School of Veterinary Medicine for helpful suggestions regarding the content and structure of the manuscript, and we are grateful to the late Dr Mary Gallagher, UCD School of Veterinary Medicine for technical assistance and mentorship of MAA.

## Author Contributions

**Conceptualization:** Katarzyna Pluta, Stephen D. Carrington, David E. MacHugh, Jane A. Irwin.

**Data curation:** Mohammed A. Alfattah, Carolina N. Correia, Paul A. McGettigan, David E. MacHugh.

**Formal analysis:** Mohammed A. Alfattah, Carolina N. Correia, Paul A. McGettigan.

**Funding acquisition:** Mohammed A. Alfattah, Carolina N. Correia.

**Investigation:** Mohammed A. Alfattah, John A. Browne, Katarzyna Pluta.

**Methodology:** Mohammed A. Alfattah, Carolina N. Correia, John A. Browne, Paul A. McGettigan, Stephen D. Carrington, Jane A. Irwin.

**Project administration:** David E. MacHugh, Jane A. Irwin.

**Resources:** Stephen D. Carrington, David E. MacHugh, Jane A. Irwin.

**Supervision:** Stephen D. Carrington, David E. MacHugh, Jane A. Irwin.

**Validation:** Mohammed A. Alfattah, Carolina N. Correia.

**Visualization:** Mohammed A. Alfattah, Carolina N. Correia.

**Writing – original draft:** Mohammed A. Alfattah, Carolina N. Correia, David E. MacHugh, Jane A. Irwin.

**Writing – review & editing:** Mohammed A. Alfattah, Carolina N. Correia, John A. Browne, Paul A. McGettigan, Katarzyna Pluta, Stephen D. Carrington, David E. MacHugh, Jane A. Irwin.

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
