## [Decision Letter · Decision Letter 0]

28 Nov 2023

PONE-D-23-31720Transcriptomics analysis of the bovine endometrium during the perioestrus periodPLOS ONE

Dear Dr. Irwin,

Thank you for submitting your manuscript to PLOS ONE. After careful consideration, we feel that it has merit but does not fully meet PLOS ONE’s publication criteria as it currently stands. Therefore, we invite you to submit a revised version of the manuscript that addresses the points raised during the review process.Please ensure that your decision is justified on PLOS ONE’s publication criteria and not, for example, on novelty or perceived impact.

We look forward to receiving your revised manuscript.

Kind regards,

Birendra Mishra, DVM, PhD

Academic Editor

PLOS ONE

5. We note that Figure(s) 4, 5, 6, 7 and 8 in your submission contain copyrighted images. All PLOS content is published under the Creative Commons Attribution License (CC BY 4.0), which means that the manuscript, images, and Supporting Information files will be freely available online, and any third party is permitted to access, download, copy, distribute, and use these materials in any way, even commercially, with proper attribution. For more information, see our copyright guidelines: http://journals.plos.org/plosone/s/licenses-and-copyright.

a. You may seek permission from the original copyright holder of Figure(s) 4, 5, 6, 7 and 8 to publish the content specifically under the CC BY 4.0 license. 

Editor Comments:

It is imperative that the resolution of the images in the figures be enhanced to meet the standards for publication. As the current images are unsuitable for publication. Furthermore, it is crucial that the manuscript be meticulously edited in compliance with the guidelines set forth by the journal.

Reviewers' comments:

Reviewer's Responses to Questions

**Comments to the Author**

1. Is the manuscript technically sound, and do the data support the conclusions?

Reviewer #1: Yes

Reviewer #2: Partly

2. Has the statistical analysis been performed appropriately and rigorously? 

Reviewer #1: Yes

Reviewer #2: Yes

3. Have the authors made all data underlying the findings in their manuscript fully available?

Reviewer #1: Yes

Reviewer #2: Yes

4. Is the manuscript presented in an intelligible fashion and written in standard English?

Reviewer #1: Yes

Reviewer #2: Yes

5. Review Comments to the Author

Reviewer #1: Excellent job with this work.

Just one question.

Did you only sample intercaruncular tissue or both caruncular and intercaruncular tissue? That information would be beneficial in the methods (ln 144-145).

Reviewer #2: Alfattah et al. (PONE-D-23-31720) performed a transcriptome analysis of estrus-synchronized heifers using RNA-seq to determine changes in gene expression during peri-estrus periods in the bovine endometrium. To extract differentially expressed genes (DEGs), analysis was based on gene expression at the onset of estrus and compared the following stages: 12 hours after CIDR removal, 24 hours after CIDR removal, 12 hours after estrus onset, 48 hours after estrus onset, and 7 days after estrus onset. They found a total of 5,845 differentially expressed genes (DEGs) from all comparisons, although the number of DEGs at 12 hours post-oestrus was low. Based on the extracted gene list, the authors attempted to identify standard pathways and functional processes of biological importance using Ingenuity Pathway Analysis, revealing pathways that are distinctive at each stage.

Overall, this is a well written paper that provides some interesting results. However, this reviewer found major issues and concerns in this manuscript.

Major Comments

The reviewer basically understands that this study is based on endometrial samples collected in the previous report (Pluta et. al, 2012, Physiol Genomics). However, the description of this manuscript does not make it clear to what extent the data are from the past and to what extent they were conducted in this study.

Pluta et. al (2012, Physiol Genomics) also performs RNA-seq. Are you using their RNA-seq data for IPA analysis?　Or are RNA extraction, library preparation and RNA-seq also performed in this study?　These should be clearly stated.

Figures and tables should be grouped together at the end of the manuscript. In the current version of the manuscript, reviewers were unable to understand the main text or legend. Are L276-278, L376-377 and L384-388 main text in Results section? It is very difficult to read.

Please provide a more detailed explanation of Figure legends for the benefit of the readers. In the current version of legends, it takes time to understand the figures and tables.

Minor Comments

L137: …according to the methods of Cooke et. al (43) and Forde et. al (20).

L246-247: What is differences in “multiple locations” and “many locations” ?

L262-268: The number of increased and decreased genes listed in parentheses should be deleted, because there were listed already in Figure 1A.

L364, 370, 378, 390 and 397: Are “Network 1” in each sections meaning the rank 1 network analyzed by IPA in each comparison? If so, should be described more clearly in the Figure legends.

L405: HDAC

L407: Remove space before “S7”.

L445: (40, 68)

L461: …by Bauersachs et. al (64) and Mitko et. al (3) ...

L566: HDAC

L953-954: Ref. 68 Is this journal, book or thesis?

6. PLOS authors have the option to publish the peer review history of their article (what does this mean?). If published, this will include your full peer review and any attached files.

Reviewer #1: No

Reviewer #2: No

---

## [Author Response · Author response to Decision Letter 0]

26 Feb 2024

Response to Reviewers

Responses to Editor’s Comments 

1. When submitting your revision, we need you to address these additional requirements. Please ensure that your manuscript meets PLOS ONE's style requirements, including those for file naming. 

We have revised our manuscript style in line with these, in particular with respect to headings, font size and file naming.

We have checked the grant numbers and provided the same grant data in the Funding Information and Financial Disclosure sections. 

The Financial Disclosure text now reads as follows:

This work was supported by a grant to MAA from the Saudi Arabian Cultural Bureau (No. IR12012/2 and IR1610; funder website https://ie.moe.gov.sa/ar/Pages/default.aspx ) and a Brazilian Science Without Borders – CAPES grant (No: BEX-13070-13-4; funder website https://www.gov.br/cnpq/pt-br/acesso-a-informacao/acoes-e-programas/programas/ciencia-sem-fronteiras ) to CNC. The funders had no role in study design, data collection and analysis, decision to publish, or preparation of the manuscript.

3. Data availability, repository information

In your Data Availability statement, you have not specified where the minimal data set underlying the results described in your manuscript can be found.

Our datasets are deposited in two locations: firstly, the raw RNA-seq data are deposited in the European Nucleotide Archive (ENA) and secondly, the bioinformatics and statistical workflow scripts are available from a public GitHub repository. Our data have now been made public.

These datasets comprise the minimal dataset underlying our work. 

The ENA data are deposited under accession number PRJEB33671.

The following text is provided at https://github.com/carolcorreia/Estrus-Endometrium-RNA-sequencing :

 Oestrus-Endometrium-RNA-sequencing

Repository containing all custom pipeline, perl scripts and R scripts used for RNA-sequencing data analyses of the Oestrus Endometrium project.

These will be published as part of a submitted publication: Alfattah, M. et al. (writing process)

In order to use the provided pipeline and scripts, user should refer first to the RNA-seq_pipeline_mohammed.txt file, which will lead to the other scripts. Further README files may be added for each scripts at later date, in order to facilitate user's utilisation and understanding.

For any inquirements concerning our article or repository, please contact:

Correia, C.N. carolina.correia [at] ucdconnect [dot] ie or MacHugh, D.E. david.machugh [at] ucd [dot] ie

We have amended our Data Availability statement to read as follows:

All of the bioinformatics and statistical workflow scripts (Bash,

Perl, and R programming languages) used are available from a public

GitHub repository https://github.com/carolcorreia/Estrus-Endometrium-RNA-sequencing

All RNA-Seq raw data used in this paper are available from the European Nucleotide Archive (ENA, https://www.ebi.ac.uk/ena/browser/search ) with accession number PRJEB33671

Copyrighted images. We note that Figure(s) 4, 5, 6, 7 and 8 in your submission contain copyrighted images. All PLOS content is published under the Creative Commons Attribution License (CC BY 4.0), which means that the manuscript, images, and Supporting Information files will be freely available online, and any third party is permitted to access, download, copy, distribute, and use these materials in any way, even commercially, with proper attribution. For more information, see our copyright guidelines: http://journals.plos.org/plosone/s/licenses-and-copyright. We require you to either (1) present written permission from the copyright holder to publish these figures specifically under the CC BY 4.0 license, or (2) remove the figures from your submission:

 You may seek permission from the original copyright holder of Figure(s) 4, 5, 6, 7 and 8 to publish the content specifically under the CC BY 4.0 license.

We have requested information from the copyright holder, TS Bioinformatics at Qiagen. This has been granted and we have received a letter from the copyright holder with permission to publish the IPA network images (Figs 4-8). This has been uploaded as part of our submission as an ‘Other’ file. The figure legends have also been amended as requested, incorporating the following text: ‘Reprinted from Qiagen IPA under a CC BY license, with permission from Qiagen, original copyright 2017’. 

5. Image resolution

We have improved the resolution of the images and have made some changes to Fig. 3 to improve legibility. The underlying graphs are the same (minus the axis labels) but were increased in size using the Photoshop image size feature with Smart Sharpen. Illustrator was then used to add editable text for the axis scales, labels, and key. The Fig 3. legend has also been expanded to make it clearer for the reader, as Reviewer 2 had concerns about the clarity of the legends.

Responses to Reviewers’ Questions:

Reviewer 1:

 ‘Did you only sample intercaruncular tissue or both caruncular and intercaruncular tissue?’. 

Both tissue types were sampled, due to a difficulty in differentiating between these at the time. To clarify this for the reader, the following line has been inserted (lines 144-145 in final version): Both caruncular and intercaruncular tissues were sampled.

Reviewer 2:

Major Comments

The reviewer basically understands that this study is based on endometrial samples collected in the previous report (Pluta et. al, 2012, Physiol Genomics). However, the description of this manuscript does not make it clear to what extent the data are from the past and to what extent they were conducted in this study.

Pluta et. al (2012, Physiol Genomics) also performs RNA-seq. Are you using their RNA-seq data for IPA analysis?　Or are RNA extraction, library preparation and RNA-seq also performed in this study?　These should be clearly stated.

The samples from the reproductive tracts of the animals were taken from the cervix and endometrium on the same date. The endometrial samples were banked and frozen at -80°C for use as part of a later project, which formed part of Mohammed Alfattah’s Ph.D. This was done separately from the work described by Pluta et al. (2012) for cervical samples. The RNA extraction from the frozen endometrial samples was done later by MAA. Library preparation, RNA-seq and bioinformatic analysis were also done by Pluta and colleagues before MAA commenced his project. Therefore, the RNA-seq data, and their analysis for this manuscript were obtained as part of a separate project and compared to the data of Pluta et al. (2012) in the Discussion section of this manuscript. The text on lines 149-150 in the final version has been amended to explain this. 

Figures and tables should be grouped together at the end of the manuscript. In the current version of the manuscript, reviewers were unable to understand the main text or legend. Are L276-278, L376-377 and L384-388 main text in Results section? It is very difficult to read.

The figure legends were embedded with the text, as this format is requested in the manuscript layout instructions for PLOS One. We agree that it can cause difficulties in interpreting the details of figures when a figure and its corresponding legend are not supplied in close proximity for easier reading.

Please provide a more detailed explanation of Figure legends for the benefit of the readers. In the current version of legends, it takes time to understand the figures and tables.

Amendments have been made to all figure legends, particularly that for Fig. 3, which has been extended significantly to provide a better explanation of what each panel in the figure means. Titles and descriptions for Tables have also been altered to provide more information for the reader. The changes made can be viewed on the marked-up copy of the manuscript. 

Minor Comments

L137: …according to the methods of Cooke et. al (43) and Forde et. al (20).

This has been amended as advised.

L246-247: What is differences in “multiple locations” and “many locations” ?

‘Multiple locations’ refers to several; ‘too many locations’ refers to multi-mapping to 10 or more locations. This has now been more clearly defined in the text. 

L262-268: The number of increased and decreased genes listed in parentheses should be deleted, because there were listed already in Figure 1A.

These have been deleted. 

L364, 370, 378, 390 and 397: Are “Network 1” in each sections meaning the rank 1 network analyzed by IPA in each comparison? If so, should be described more clearly in the Figure legends.

This is correct. See amended figure legend for details.

L405: HDAC

This has been changed on this line, and also where it appears elsewhere, from Hdac to HDAC.

L407: Remove space before “S7”.

This has been done.

L445: (40, 68)

This has been changed to ‘by Pluta et al. [40] and Pluta [68]’, now on line 473 in manuscript.

L461: …by Bauersachs et. al (64) and Mitko et. al (3) ...

This has been changed as requested.

L566: HDAC See above.

L953-954: Ref. 68 Is this journal, book or thesis? 

This reference refers to Kasia Pluta’s UCD Ph.D thesis (2011) and this detail has now been added to the Reference list.

---

## [Decision Letter · Decision Letter 1]

11 Mar 2024

Transcriptomics analysis of the bovine endometrium during the perioestrus period

PONE-D-23-31720R1

Dear Dr. Irwin,

We’re pleased to inform you that your manuscript has been judged scientifically suitable for publication and will be formally accepted for publication once it meets all outstanding technical requirements.

Kind regards,

Birendra Mishra, DVM, PhD

Academic Editor

PLOS ONE

Additional Editor Comments (optional):

All comments have been addressed.

Reviewers' comments:

Reviewer's Responses to Questions

**Comments to the Author**

1. If the authors have adequately addressed your comments raised in a previous round of review and you feel that this manuscript is now acceptable for publication, you may indicate that here to bypass the “Comments to the Author” section, enter your conflict of interest statement in the “Confidential to Editor” section, and submit your "Accept" recommendation.

Reviewer #1: All comments have been addressed

Reviewer #2: All comments have been addressed

2. Is the manuscript technically sound, and do the data support the conclusions?

Reviewer #1: Yes

Reviewer #2: Yes

3. Has the statistical analysis been performed appropriately and rigorously? 

Reviewer #1: Yes

Reviewer #2: Yes

4. Have the authors made all data underlying the findings in their manuscript fully available?

Reviewer #1: Yes

Reviewer #2: Yes

5. Is the manuscript presented in an intelligible fashion and written in standard English?

Reviewer #1: Yes

Reviewer #2: Yes

6. Review Comments to the Author

Reviewer #1: (No Response)

Reviewer #2: This manuscript, Transcriptomics analysis of the bovine endometrium during the perioestrus period, has been revised and resubmitted to PLOS ONE for re-evaluation.

This reviewer sees the improvement and rather appreciates their efforts in their revision. My original concerns have been solved.

7. PLOS authors have the option to publish the peer review history of their article (what does this mean?). If published, this will include your full peer review and any attached files.

Reviewer #1: No

Reviewer #2: No

---

## [Editor Report · Acceptance letter]

19 Mar 2024

PONE-D-23-31720R1 

PLOS ONE

Dear Dr. Irwin, 

I'm pleased to inform you that your manuscript has been deemed suitable for publication in PLOS ONE. Congratulations! Your manuscript is now being handed over to our production team.

Kind regards, 

on behalf of

Dr. Birendra Mishra 

Academic Editor

PLOS ONE